# Stepwise modifications of transcriptional hubs link pioneer factor activity to a burst of transcription

Chun-Yi Cho [ORCID][1] & Patrick H. O'Farrell [ORCID][1] ✉

Binding of transcription factors (TFs) promotes the subsequent recruitment of coactivators and preinitiation complexes to initiate eukaryotic transcription, but this time course is usually not visualized. It is commonly assumed that recruited factors eventually co-reside in a higher-order structure, allowing distantly bound TFs to activate transcription at core promoters. We use live imaging of endogenously tagged proteins, including the pioneer TF Zelda, the coactivator dBrd4, and RNA polymerase II (RNAPII), to define a cascade of events upstream of transcriptional initiation in early *Drosophila* embryos. These factors are sequentially and transiently recruited to discrete clusters during activation of non-histone genes. Zelda and the acetyltransferase dCBP nucleate dBrd4 clusters, which then trigger pre-transcriptional clustering of RNAPII. Subsequent transcriptional elongation disperses clusters of dBrd4 and RNAPII. Our results suggest that activation of transcription by eukaryotic TFs involves a succession of distinct biomolecular condensates that culminates in a self-limiting burst of transcription.

In eukaryotes, the recruitment of RNA polymerase II (RNAPII) to transcription start sites on DNA depends on the assembly of the pre-initiation complex (PIC) and is regulated by hundreds of trans-acting factors[1,2]. In particular, transcription factors (TFs) recruit nucleosome remodelers, histone modifiers, and Mediator to promote the formation of PIC. How these numerous upstream inputs are integrated to give the extraordinary specificity and intricacy of transcriptional regulation remains incompletely understood. A common view suggested by biochemical studies is that these factors are progressively assembled into a single final complex through cooperative interactions. However, other sophisticated processes initiating DNA replication and promoting splicing of mRNAs are governed by a series of distinct and ephemeral complexes in which each complex promotes the next in energy-driven steps[3]. Here, we are interested in the possibility that initiation of transcription similarly involves directional transformations of intermediate complexes that would provide additional opportunity for specificity and regulation.

Visualizing the composition of transcriptional machinery over time might detect intermediate complexes that integrate the multitude of regulatory inputs of transcriptional control. In recent years, advances in confocal and super-resolution imaging led to the discovery that a wide variety of transcriptional regulators are recruited to form clusters at active genes[4]. These clusters are thought to function as "transcriptional hubs" by locally enriching transcriptional machinery and enhancing their binding to target DNA sites. Transcriptional hubs are a type of membraneless compartment, whose formation typically involves the multivalent interaction between intrinsically disordered regions (IDRs)[5]. Accordingly, IDRs are commonly found in the activation domains of TFs as well as the C-terminal domain (CTD) of Rpb1 in RNAPII[6]. Similar to the idea that a single final complex is assembled on the DNA to initiate transcription, it has been proposed that the heterotypic interactions between IDRs can give rise to a compartment that simultaneously enriches TFs, coactivators, Mediator, and RNAPII at promoters[7]. Nonetheless, how transcriptional hubs are regulated and whether they undergo compositional changes are still unclear.

Studying the dynamics of transcriptional hubs in living cells is complicated by the discontinuous and stochastic nature of eukaryotic transcription, a phenomenon also known as bursting[8]. The *Drosophila* embryo provides a powerful context to study the timing of events

[1]Department of Biochemistry and Biophysics, University of California, San Francisco, San Francisco, CA 94158, USA. ✉e-mail: ofarrell@cgl.ucsf.edu

upstream of transcriptional initiation. The early wave of transcription in *Drosophila* embryos is coupled to the rapid nuclear division cycles such that a few hundred genes initiate a burst of transcription about 3 min after each mitosis[9–11]. The synchrony of early nuclear cycles and real-time localization of tagged proteins allow one to track activation events prior to the onset of transcription, and tools to knockdown function are available to assess the contribution of events to gene activation. In a recent study, live imaging of endogenously tagged RNAPII revealed the abrupt appearance of RNAPII clusters 2–3 min after mitosis[12]. Brief metabolic labelling revealed foci of nascent transcripts throughout the nuclei in fixed embryos—these foci broadly colocalized with RNAPII clusters, indicating that early-forming RNAPII clusters mark sites of active transcription. Importantly, as nascent transcript levels increased, RNAPII clusters declined and eventually dispersed. These observations are consistent with numerous observations[12–15] and support a model in which a large excess of RNAPII is recruited prior to initiation, which is then inefficiently converted to elongating RNAPII. What produces this pre-transcriptional RNAPII clustering and how it is coordinated with a burst of transcription are not yet fully understood. Here, we follow events during the ~2.5 min between mitotic exit and the formation of RNAPII clusters and the fate of these clusters as transcription ensues at about 3 min after mitosis.

Zelda (Zld) is a pioneer TF that widely promotes the early wave of zygotic gene expression[16–18]. Maternally supplied Zld binds to thousands of enhancers and promoters, and its binding sites exhibit increased chromatin accessibility and histone acetylation[19–24]. Depletion of maternally expressed Zld curtails early zygotic transcription, and the embryos become highly defective at the mid-blastula transition (MBT)[16]. The transactivation domain of Zld has been mapped to an intrinsically disordered region[25]. Moreover, fluorescently tagged Zld forms highly dynamic clusters in the nucleus[26,27], and previous studies suggest that Zld clusters increase the local concentration of other TFs and facilitate their binding to target DNA[26,28,29]. Knockdown of Zld reduces RNAPII "speckles" in fixed embryos[30]. While these previous studies support a model in which Zld promotes the recruitment of additional components to form transcriptional hubs and facilitates the onset of zygotic transcription, the exact mechanism has not been determined.

In this study, we combine genetic perturbation and real-time imaging to delineate a pathway that nucleates and serially transforms transcriptional hubs to trigger initiation of transcription in early *Drosophila* embryos. We show that Zld acts through transcription coactivators, including the lysine acetyltransferase dCBP and the BET protein dBrd4, to initiate RNAPII clustering at non-histone genes. Importantly, real-time imaging reveals only limited colocalization of these factors at transcriptional hubs, suggesting dynamic and directional changes in the composition such that upstream activators do not stably persist in the hubs with downstream effectors and RNAPII. We propose a model in which Zld forms numerous largely unstable clusters, some of which trigger a dCBP-dependent step to build more stable dBrd4 clusters; a subset of these dBrd4 clusters then promotes RNAPII clustering near active promoters, and this pool of RNAPII fuels a burst of transcription. Inhibition of transcriptional elongation stabilizes some Zld and dBrd4 clusters, indicating that transcription directly or indirectly promotes their dispersal. Finally, while early inhibition of transcription inhibits RNAPII clustering, abrupt inhibition of transcript elongation after hub formation stabilizes RNAPII clusters. These findings indicate that transcription destabilizes hubs, a feedback that could lead to cycles of RNAPII accumulation and depletion, thereby contributing to the busting feature of transcription. We suggest that the onset of transcription, like the onset of replication, involves upstream events that directionally modify the machinery to precisely control the process.

## Results

### The pioneer transcription factor Zelda acts with the lysine acetyltransferase dCBP to initiate RNAPII clustering

We sought to understand what triggers the abrupt formation and subsequent dispersal of RNAPII clusters during a burst of transcription in early *Drosophila* embryos[12]. As a previous study showed that the depletion of Zld reduced RNAPII speckles in immunostaining[30], we wanted to further characterize this process using real-time approaches. To block the actions of Zld in the nucleus, we sequestered endogenously GFP-tagged Zld in the cytoplasm by the JabbaTrap, which is an anti-GFP nanobody fused with the lipid-droplet protein Jabba, and then recorded RNAPII dynamics using mCherry-tagged Rpb1[12,31,32]. During a normal cell cycle in control embryos, RNAPII abruptly formed two classes of clusters about 2–3 min after mitosis, including the large clusters at the two histone locus bodies (HLBs) and more numerous small clusters (Fig. 1a, top). In embryos injected with JabbaTrap mRNA prior to nuclear cycle 12, inhibition of GFP-tagged Zld in cycle 12 blocked the formation of small RNAPII clusters at non-histone genes but not the large clusters at HLBs (Fig. 1a, bottom). The timing of injection of *JabbaTrap* mRNA can be adjusted, and because the accumulated JabbaTrap sequesters its nuclear targets during mitosis when nuclear membrane breakdown exposes Zld to the cytoplasmic trap, we achieved abrupt trapping of Zld at the transition from one cycle to the next. We found that abrupt sequestration of Zld in mitosis 12 blocked most RNAPII clustering in the following interphase in cycle 13 (Supplementary Fig. 1). Thus, Zld is required in both cycle 12 and 13 to initiate RNAPII clustering at non-histone genes, consistent with a role of Zld in accelerating transcription after mitosis[27,28].

Since Zld is required for RNAPII clustering at non-histone genes, we explored the spatial and temporal localization of Zld and RNAPII. It has been shown that Zld forms highly dynamic and transient clusters in the interphase nucleus and that both Zld and RNAPII clusters are the most prominent before substantial accumulation of nascent transcripts[12,26,27]. It was previously concluded that "long-lasting stable contacts between sites of transcription and Zelda dense regions are not detected"[26]. In a temporal sequence preceding the onset of transcription in cycle 12 (Fig. 1b), two features are clear: there are many more Zld clusters than RNAPII, and the Zld clusters appear earlier than the RNAPII clusters. Temporally, Zld clusters emerged within 60 s post mitosis, while the RNAPII signal was gradually but uniformly accumulating in the nucleus. RNAPII clusters began to form at 120 s post mitosis and matured at 240 s when there was no obvious Zld colocalizing with RNAPII clusters (Fig. 1b, c). We conclude that Zld and RNAPII do not stably comingle in clusters and that most Zld clusters do not trigger formation of RNAPII clusters. These observations are not consistent with a simple model of deterministic and sequential recruitment by direct interaction.

We wanted to test whether a subset of more stable Zld clusters might act in a direct but transient manner to nucleate RNAPII clusters. This proved to be difficult to resolve. Zld clusters were too numerous and heterogeneous for us to identify individually, and they were too unstable or motile to track. Nonetheless, we looked over time to see whether newly forming RNAPII clusters were associated with Zld clusters. While there are a few tantalizing associations (Fig. 1b, boxed areas), they have unexpected features. The RNAPII clusters formed adjacent to the Zld clusters, and there are dynamic shifts in the relationship of the two signals with marked disappearance of Zld as the RNAPII clusters grew. The findings do not discount the possibility of some direct interaction between Zld and RNAPII, nor do they rigorously document that such interactions occur. However, the absence of persistent and co-extensive colocalization of the two proteins led us to hypothesize that Zld acts indirectly though intermediate steps. We reasoned that identifying the cofactors might provide insights into the mode of Zld actions in the formation of RNAPII clusters.

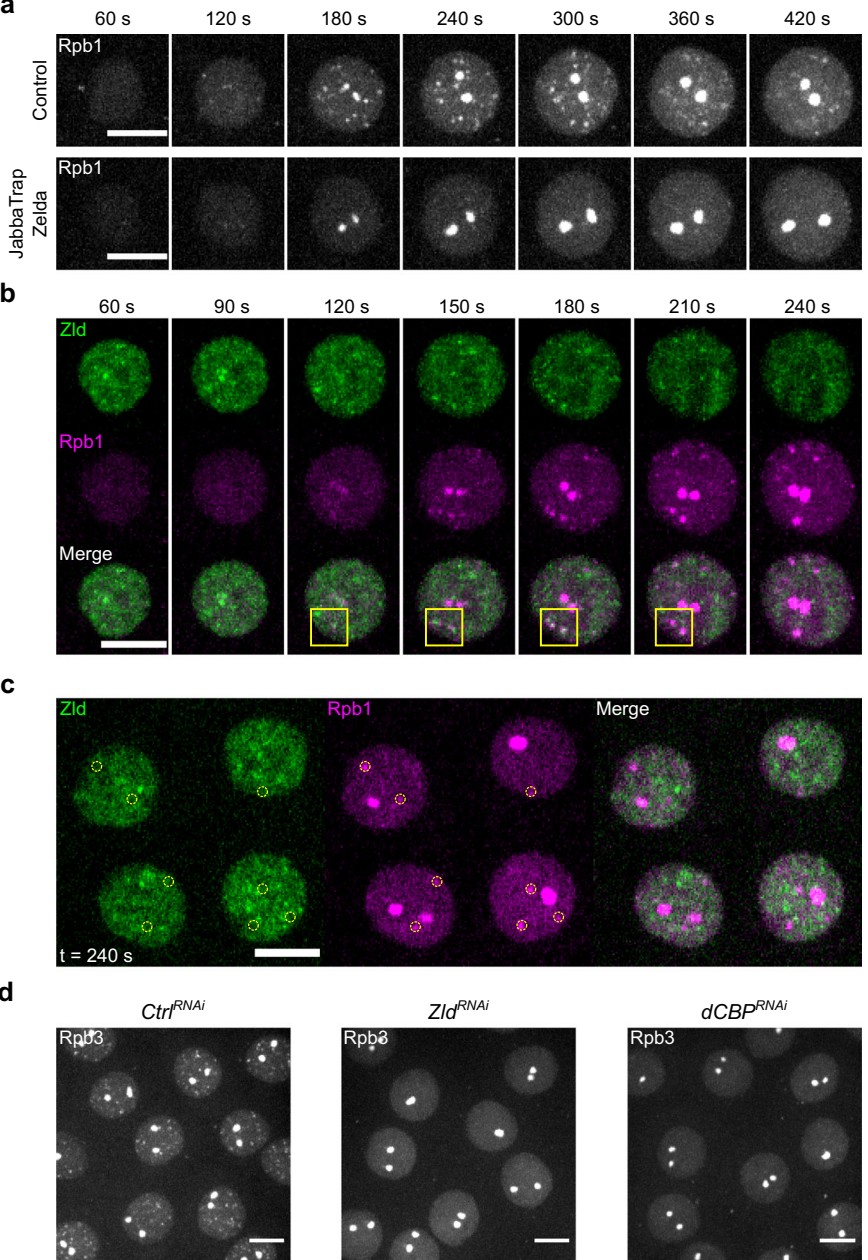

**Fig. 1 | The pioneer transcription factor Zelda acts with the lysine acetyltransferase dCBP to initiate clustering of RNAPII at non-histone genes.** **a** Representative stills from live imaging of mCherry-Rpb1 in cycle 12 embryos either untreated (control) or in which sfGFP-Zld is trapped in the cytoplasm, thereby depleting Zld function. Embryos were injected with either water (top) or JabbaTrap mRNA (bottom), whose protein product sequestered endogenously GFP-tagged Zld to lipid droplets (see Supplementary Fig. 1 for example). Time in interphase 12 is indicated. Similar outcomes were observed in 9 embryos for each treatment. **b** Representative stills from live imaging of sfGFP-Zld and mCherry-Rpb1 during cycle 12. Time in interphase 12 is indicated. Yellow squares show cases of close approximations of the Zld and Rpb1 signals discussed in the text. Similar outcomes were observed in 3 embryos. **c** Single-z-plane images from live imaging of sfGFP-Zld and mCherry-Rpb1 as described in (**b**) at 4 min in interphase 12. Yellow circles mark the positions of small RNAPII clusters, which did not show colocalization with Zld clusters. **d** Snapshots from live imaging of EGFP-Rpb3 in embryos expressing shRNA targeting *white* (control), *zld*, or *dCBP*. Frames at 3–4 min in interphase 12 are displayed. Similar outcomes were observed in at least 5 embryos for each RNAi. All scale bars in Fig. 1 indicate 5 μm. Maximal projections are shown unless otherwise noted.

Previous work suggested events that might mediate an indirect requirement of Zld for RNAPII recruitment. Zld binding increases chromatin acetylation and DNA accessibility[19–23], but the coactivators that act with Zld remain unknown. Among the chromatin changes associated with Zld, the acetylation of histone H3 lysine 27 (H3K27ac) is known to be deposited by the lysine acetyltransferase CBP/p300 and is a major determinant of enhancer activity[23,24,33]. Upon testing candidates, we found that the *Drosophila* homolog of CBP, encoded by *Nejire* (*Nej*), is necessary for RNAPII clustering. Because dCBP/Nej is required for the production of eggs[34,35], we used RNAi to deplete maternal dCBP in early embryos by expressing shRNA targeting dCBP only from the late stages of oogenesis (Supplementary Fig. 2a)[36,37]. Immunostaining confirmed the global reduction of H3K27ac in dCBP-knockdown embryos (Supplementary Fig. 2b), consistent with a recent finding that dCBP is the major acetyltransferase responsible of H3K27 acetylation in the early embryo[38]. Except for the large HLBs, the

clustering of EGFP-tagged Rpb3 (another subunit of RNAPII) in cycle 12 was blocked by the knockdown of dCBP, phenocopying the knockdown of Zld (Fig. 1d)[21]. Knockdown of dCBP did not reduce the clustering of mNeonGreen-tagged Zld (Supplementary Fig. 2c, d)[26], showing that dCBP acts at a downstream step after the formation of Zld clusters.

We conclude that Zld and dCBP are both required to initiate RNAPII clustering at non-histone genes and suggest that dCBP either acts as a cofactor for Zld action or is a downstream factor activated by Zld to mediate RNAPII clustering.

## The BET protein dBrd4 acts downstream of Zld and dCBP to initiate RNAPII clustering

We hypothesized that acetylation marks deposited by dCBP subsequently initiate RNAPII clustering via recruitment of chromatin readers, such as those recognizing acetylated lysine residues. In line with this hypothesis, the mammalian bromodomain and extraterminal (BET) protein Brd4 binds acetylated histones and is critical for the formation of transcriptional condensates at super-enhancers[39–41]. *Drosophila melanogaster* has only one locus, *female sterile (1) homeotic*, encoding 2 isoforms of a BET homolog, dBrd4 herein. Like its mammalian orthologs, dBrd4 functions in transcriptional regulation, as shown in cell lines and gastrulating embryos[42–45]. However, the role of maternally supplied dBrd4 in the minor wave of zygotic transcription in early embryos has not been examined, partly due to the female sterility of genetic mutants[46].

We used CRISPR-Cas9 to tag the N-terminus of endogenous dBrd4 with either HaloTag or sfGFP. The N-terminal tag is present in both the long and short protein isoforms of dBrd4, as confirmed by western blot of sfGFP-dBrd4 from pre-MBT embryos (Supplementary Fig. 3a). Flies homozygous for HaloTag-dBrd4 or sfGFP-dBrd4 are healthy and fertile, and they lay eggs that hatch at a rate similar to wild type (Supplementary Fig. 3b). We then used confocal live imaging to examine the localization of tagged dBrd4 in early embryos. Using either TMR-HaloTag-dBrd4 or sfGFP-dBrd4, we observed the broad association of dBrd4 with chromosomes (Supplementary Fig. 3c), similar to previous observations and the behaviors of Brd4 homologs in other systems[41,47–49]. In addition to the general chromatin interaction with mitotic chromosomes, we observed discrete clusters of dBrd4 during most of the cell cycle except a short period at the beginning of interphase (e.g., Fig. 2a). We set out to understand the regulation and function of these clusters.

To test the hypothesis that dBrd4 acts downstream of Zld and dCBP, we first used RNAi to knockdown Zld or dCBP and then examined dBrd4 localization in cycle 12. In control embryos, numerous dBrd4 clusters were visible by 3 min after mitosis (Fig. 2a, b). Following the depletion of Zld, only one to two dBrd4 clusters emerged and intensified in each nucleus after mitosis (Fig. 2a, frames in interphase; Fig. 2b). These residual dBrd4 clusters developed a signal similar to the HLBs, which normally became the dominant sites of dBrd4 recruitment in late interphase and mitosis (Fig. 2a, frames in mitosis; Supplementary Fig. 4). Thus, the formation of non-HLB dBrd4 clusters depends on Zld, and it appears that an alternative pathway can promote dBrd4 recruitment to HLBs.

The depletion of dCBP severely disrupted the formation of all dBrd4 clusters, suggesting that dCBP is required in both the Zld-dependent pathway for the formation of the majority of the dBrd4 clusters, as well as in the Zld-independent pathway at HLBs. Note that in addition to impairing cluster formation in interphase, the association of dBrd4 with mitotic chromosomes was reduced at non-HLB positions by Zld knockdown and was ubiquitously reduced by dCBP knockdown (Fig. 2a, frames in mitosis). We conclude that Zld and another HLB-specific pathway regulate the function of dCBP, which is the major acetyltransferase required for dBrd4 clustering in early embryos.

We next asked whether dBrd4 is required for RNAPII clustering. We attempted to inhibit endogenous sfGFP-dBrd4 in early embryos by maternally expressing JabbaTrap using the Gal4/UASp system; however, we found that the combination of sfGFP-dBrd4 and JabbaTrap germline expression led to female sterility, consistent with the genetic mutant phenotype[46,50]. We thus maternally expressed *JabbaTrap* mRNA with a *bicoid* 3'UTR, which, in addition to sequences directing anterior localization of the mRNA, includes sequences that suppress translation prior to fertilization. This allowed us to collect embryos in which sfGFP-dBrd4 was maternally expressed but was sequestered in the cytoplasm after fertilization. Notably, while the anterior localization of mRNA by *bicoid* 3'UTR might create a gradient in the concentration of JabbaTrap, we observed sequestration of sfGFP-dBrd4 throughout the embryos (Supplementary Fig. 5a). In these embryos, the formation of RNAPII clusters was fully blocked except for the large clusters at HLBs (Fig. 2c). Similarly, displacing dBrd4 from chromatin using the bromodomain inhibitor JQ1 selectively blocked the formation of small RNAPII clusters but not those at HLBs (Supplementary Fig. 5b)[49,51,52]. Finally, the dBrd4 JabbaTrap embryos were lethal and did not complete cellularization and gastrulation at the MBT (Fig. 2d), phenocopying the *zld*⁻ null mutant and suggesting global impairment of the early zygotic transcriptional program[16].

We conclude that dBrd4 is a required effector of Zld and dCBP for initiating RNAPII clustering at non-histone genes.

## Clusters of Zld and dBrd4 emerge sequentially in interphase with little indication of colocalization

The above data suggest a model in which the transient binding of Zld to target DNA promotes localized acetylation by dCBP, which at some sites reaches threshold levels that nucleate dBrd4 clusters, which subsequently promote pre-transcriptional RNAPII clustering near promoters. Accordingly, we would expect the dBrd4 clusters to form only after the emergence of Zld clusters. To test this, we performed simultaneous live imaging of mNeonGreen-Zld and TMR-HaloTag-dBrd4 in cycle 12 (Fig. 3a).

Upon exiting mitosis 11, Zld rapidly bound to telophase chromosomes and began to form clusters in the nucleus within a minute in interphase; in contrast, during this first minute, the mitotic clusters of dBrd4 diminished as chromatin became decompacted, and most of these dBrd4 clusters were dispersed (Fig. 3a, 0–60 s). Around 2 min into interphase, dBrd4 began to form clusters anew, and then both Zld and dBrd4 clusters remained detectable throughout the rest of interphase (Fig. 3a).

We sought to detail this temporal sequence. However, as previously described, Zld clusters are abundant and short lived[26,27], making it impossible for us to track the clusters from one frame to the next at our imaging speed. Consequently, we were unable to follow individual Zld clusters to see if and when they might recruit dBrdr4. We looked for a quantitative measure of global clustering to compare the timing of Zld to dBdr4 clustering. The large number of Zld clusters and their varied intensities made precise enumeration of clusters and their intensities impractical. Since clustering transforms nuclear intensity from a homogeneous into a heterogenous distribution, it leads to an increase in the variance of recorded intensity signals. Hence, we have used variance of pixel intensities within the nucleus as a simple metric of clustering. We measured this variance in all the recorded nuclei obtained from imaging of three embryos, and we plotted the mean-variance versus time to gain a global view of the timing of clustering. The variance of the Zld signal was already high at 30 s post mitosis, peaked at 60 s, and then declined gradually. The variance of the dBrd4 signal increased later with a lag longer than 1 min compared to Zld (Fig. 3b). We conclude that Zld and dBrd4 clusters emerge with a defined temporal order at the beginning of interphase.

Since most models for the formation of transcriptional hubs suggest cooperative assemblies of proteins, they would predict

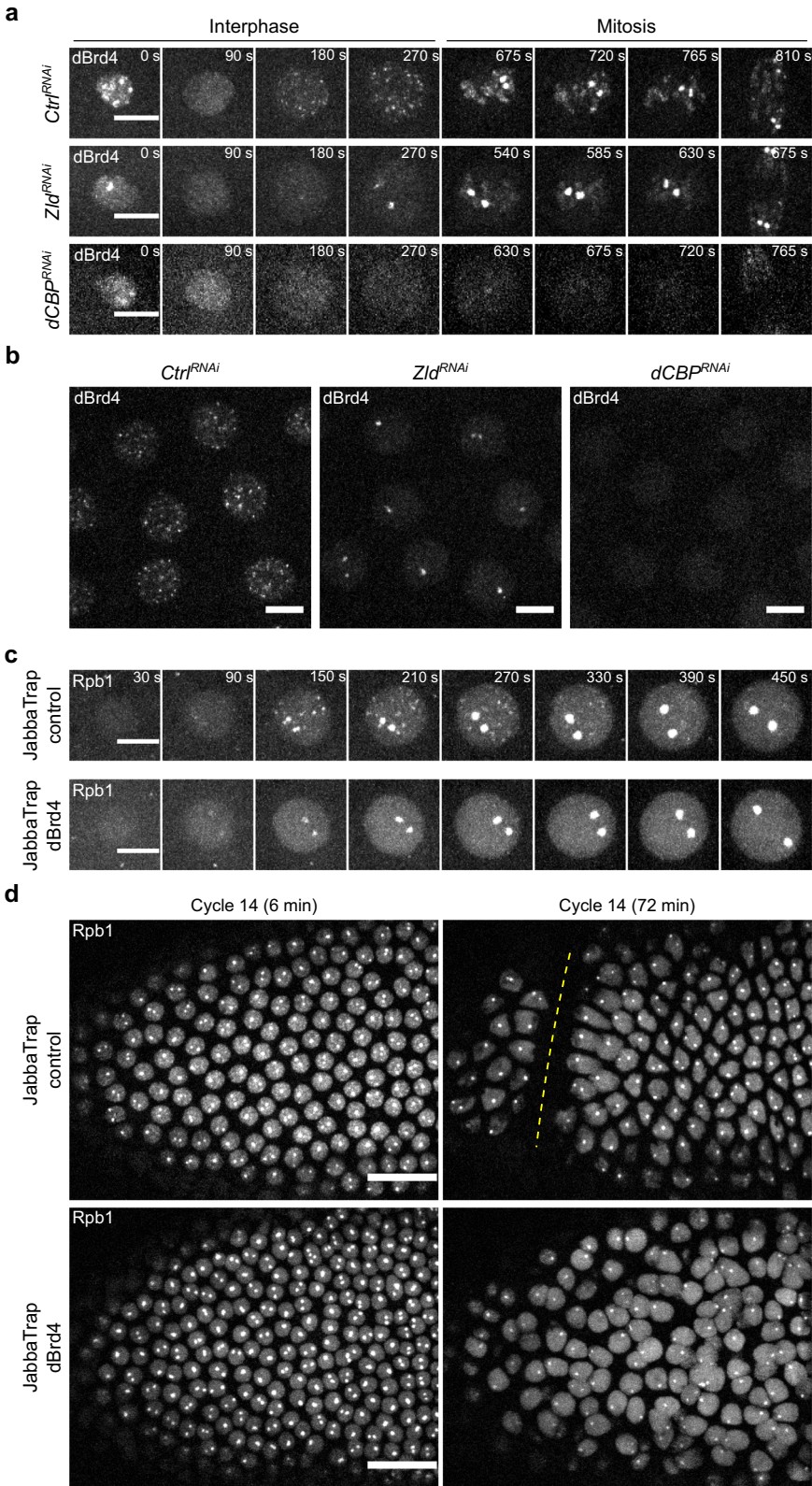

colocalization of components. We visually inspected single-z-plane images more carefully to determine the extent to which Zld and dBrd4 clusters colocalized. We found that dBrd4 clusters did not overlap extensively with Zld clusters either when they initially formed (Fig. 3c) or when they became more prominent at later times (Fig. 3d). While we cannot confidently exclude the possibility of very transient colocalization, we see no convincing evidence for it. Whether or not Zld and dBrd4 very transiently colocalize, our findings are consistent with the model that after recruiting dCBP to increase local acetylation level, Zld is no longer needed to sustain dBrd4 clusters, and we observe no perduring presence of Zld in the dBrd4 clusters.

### The spatiotemporal relationship between dBrd4 and RNAPII clusters

Next, we performed simultaneous live imaging of sfGFP-dBrd4 and mCherry-Rpb1 (Fig. 4a). We found that the emergence and maturation

**Fig. 2 | The BET protein dBrd4 acts downstream of Zld and dCBP to initiate RNAPII clustering. a** Representative stills from live imaging of sfGFP-dBrd4 in embryos derived from mothers expressing an shRNA in their germline targeting either *white* (control), *Zld*, or *dCBP*. Time in interphase 12 is indicated. In order to detect anything, the brightness and contrast for images of *dCBP* RNAi embryos were enhanced relatively to the control and *Zld* RNAi embryos. Similar outcomes were observed in at least 5 embryos for each RNAi. **b** Snapshots of a field from the movies described in (**a**). Frames at 3–4 min in interphase 12 are displayed. **c** Representative stills from live imaging of mCherry-Rpb1 during cycle 12. Both the JabbaTrap-control and JabbaTrap-dBrd4 embryos were from females expressing

*JabbaTrap* mRNA with *bicoid* 3'UTR in the germline. The untagged dBrd4 in the control embryo was unaffected by JabbaTrap, whereas endogenously GFP-tagged dBrd4 was sequestered in the cytoplasm by JabbaTrap in the JabbaTrap dBrd4 embryos. See Methods and Supplementary Table 2 for additional details. Similar outcomes were observed in 6 embryos. **d** Representative stills from live imaging of mCherry-Rpb1 during cycle 14 showing the effect of JabbaTrap as control embryos went through cellularization and gastrulation. Anterior poles are to the left. The cephalic furrow is marked by dashed line. The genotypes are as described in (**c**). Similar outcomes were observed in 3 embryos. Scale bars in **a**–**c**, 5 µm; scale bars in **d**, 20 µm. All images shown are maximal projections.

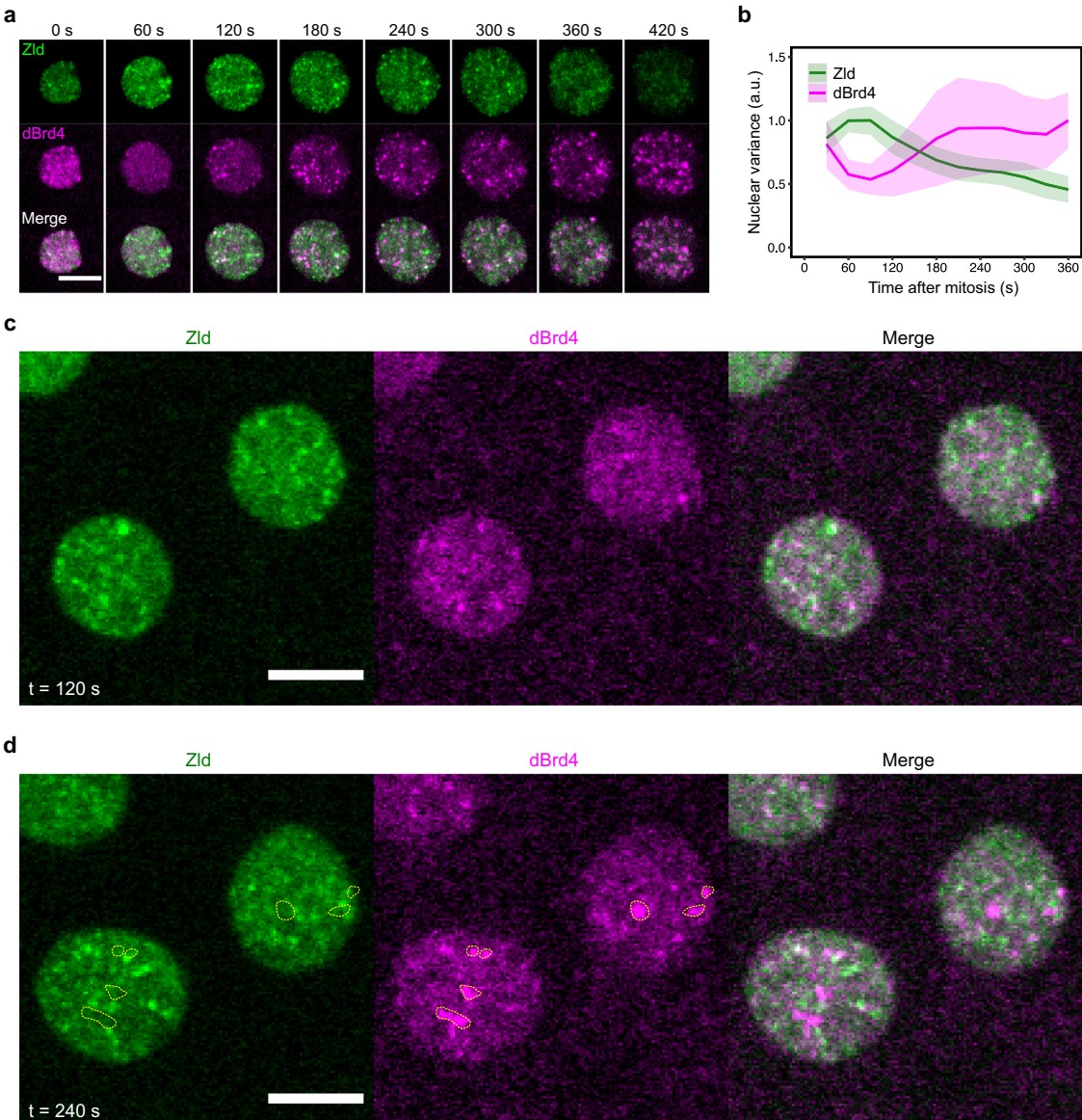

**Fig. 3 | Interphase clusters of Zld and dBrd4 emerge sequentially in interphase with little indications of colocalization. a** Representative stills from live imaging of mNeonGreen-Zld and TMR-HaloTag-dBrd4 during cycle 12. Maximal projections are shown. Time in interphase 12 is indicated. **b** Variance of fluorescent intensities for tagged Zld or dBrd4 in the nucleus during cycle 12. The mean values of data

from 3 embryos are presented. Shaded areas represent SD. A total of 101 nuclei were recorded and analyzed. **c**, **d** Single z-plane images from movies as described in (**a**). Time in interphase 12 is indicated. dBrd4 clusters that do not overlap with Zld clusters are circled in yellow. All scale bars in Fig. 3 indicate 5 µm. Source data are provided as a Source Data file.

of RNAPII clusters were delayed for about 1 min compared to dBrd4 (Fig. 4a, b). Thus, clusters of dBrd4 and RNAPII also emerged sequentially with a defined temporal order in interphase.

In contrast to Zld clusters, dBrd4 clusters are less numerous and more stable, allowing us to track the more prominent clusters with

some confidence during the time when RNAPII clusters begin to form (Fig. 4c, d; Supplementary Movie 1). Here, we examined cycle 11 when there were fewer clusters as compared to cycle 12, allowing easier tracking. In a typical example, we observed the initial emergence of a faint RNAPII cluster within an already bright dBrd4 cluster (Fig. 4c, 0 s;

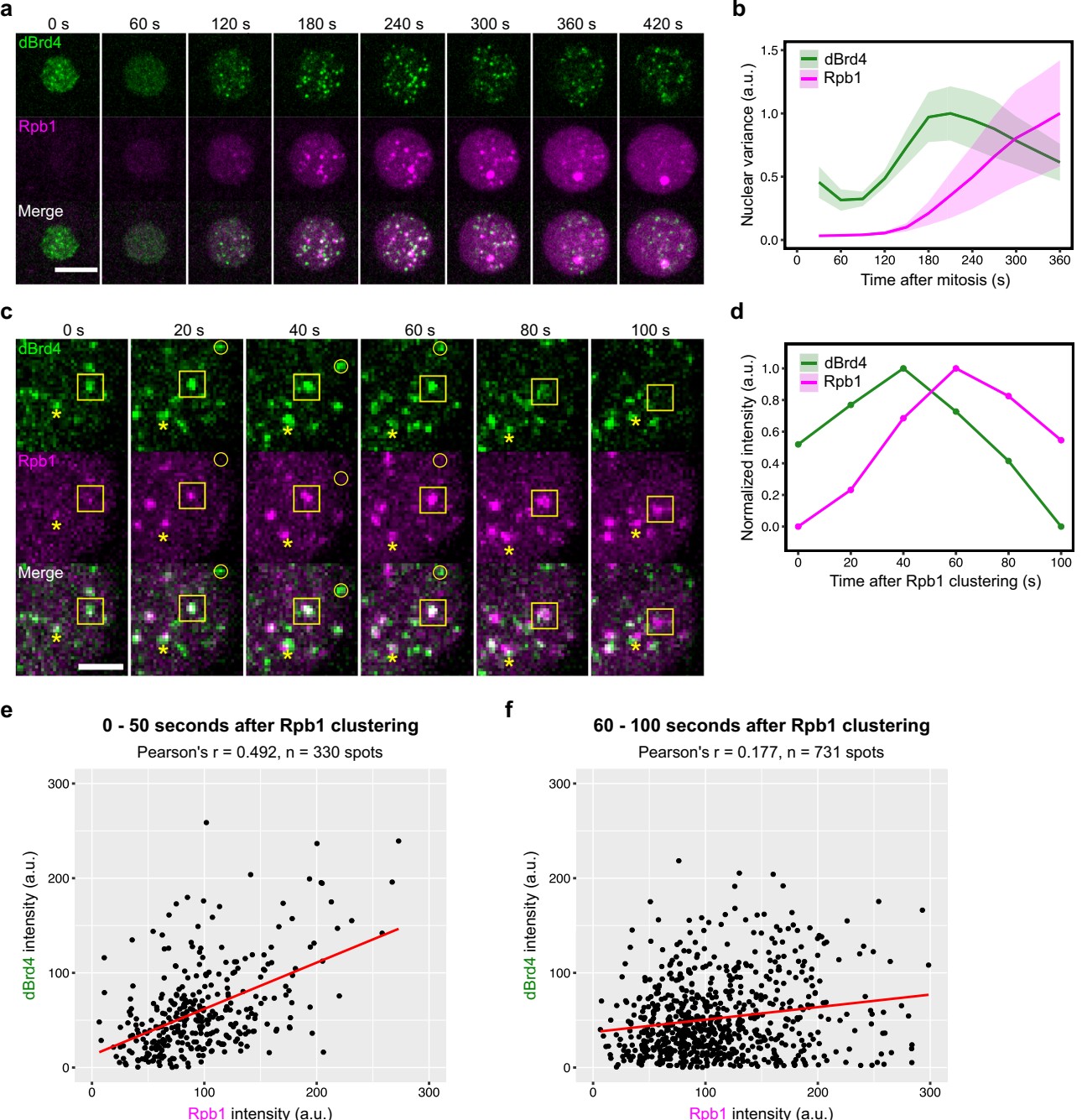

**Fig. 4 | Co-clusters of dBrd4 and RNAPII undergo maturation and dispersal with a temporal delay. a** Representative stills from live imaging of sfGFP-dBrd4 and mCherry-Rpb1 during cycle 12. Maximal projections are shown. Time in interphase 12 is indicated. **b** Variance of fluorescent intensities for tagged dBrd4 or Rpb1 in nuclei during cycle 12. The mean values of data from 3 embryos are presented. Shaded areas represent SD. A total of 84 nuclei were recorded and analyzed. **c** Zoomed-in stills from live imaging of sfGFP-dBrd4 and mCherry-Rpb1 during cycle 11. Maximal projections of a portion of a nucleus are shown at a 20 s frame rate. Time relative to the earliest detection of RNAPII clusters in this nucleus is indicated. The box tracks a pair of dBrd4 and RNAPII clusters as they sequentially emerge and disperse in the movie. The asterisk marks another locus that recruits both dBrd4 and RNAPII. The circle indicates a locus or perhaps different ephemeral but nearby loci where dBrd4 clustering is not followed by RNAPII recruitment. Scale bar, 2 μm. **d** Normalized intensities of dBrd4 and Rpb1 at the cluster outlined in the yellow box in **c**. **e**, **f** Scatter plots showing the integrated intensity of Rpb1 and dBrd4 at spots near Rpb1 clusters. Data are shown for the indicated time periods shortly after the appearance of Rpb1 clusters (**e**) or at later times (**f**). Red line represent the line of best fit by linear regression. Movies of 3 embryos including that shown in **c** were analyzed. a.u., arbitrary unit. Source data are provided as a Source Data file.

yellow box). The pair of dBrd4 and RNAPII clusters briefly (~40 s) increased in intensity jointly, with the RNAPII cluster enlarging to essentially fill the dBdr4 cluster. The initially near coextensive overlap of dBrd4 and RNAPII clusters suggests the comingling of the proteins, which might involve direct protein-protein interactions or joint partitioning into a common coacervate.

Some dBrd4 clusters emerged and disappeared without recruiting RNAPII (Fig. 4c, circles). This argues that locally high dBrd4 levels are not sufficient to trigger RNAPII clustering. This suggests that some difference, such as the presence of activators or absence of repressors, guides some but not all of the dBrd4 clusters to recruit RNAPII clusters. In one view, the dBrd4 clusters that vanish without producing a RNAPII

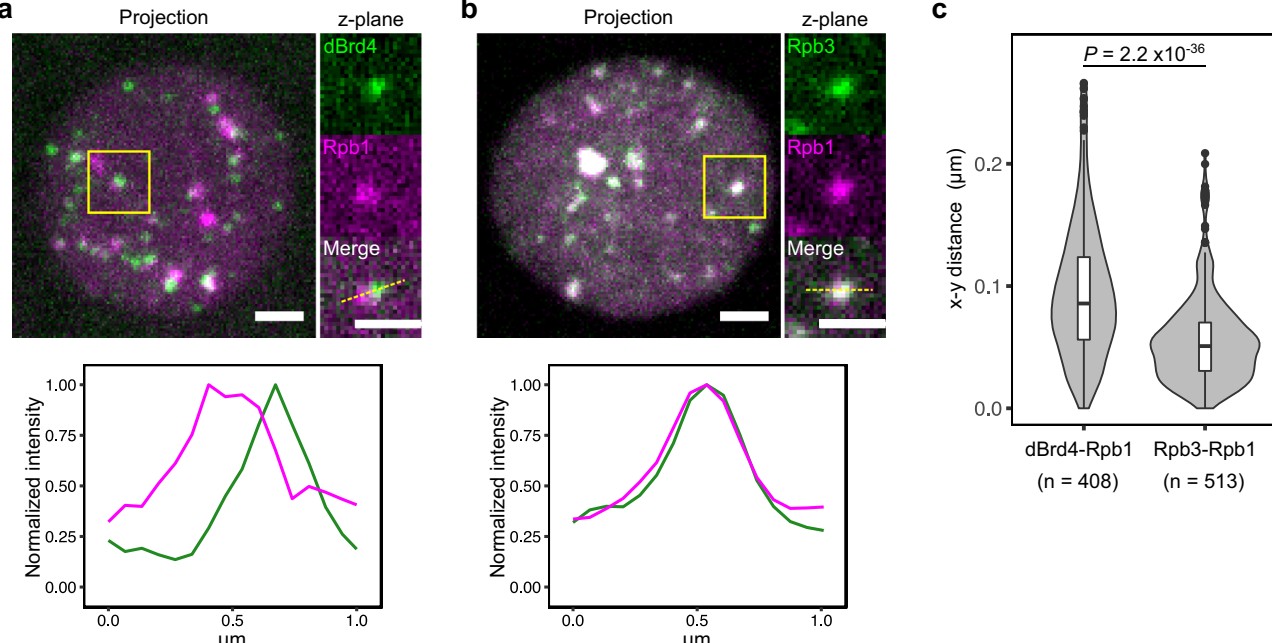

**Fig. 5 | dBrd4 and RNAPII do not always co-mingle in a single cluster but are adjacent or overlapping. a** Snapshots from imaging of sfGFP-dBrd4 and mCherry-Rpb1 in an embryo injected with 2% formaldehyde and fixed for 8 min (top). The injection of formaldehyde was performed at about 3 min in cycle 12 to fix the clusters of dBrd4 and Rpb1. Intensity profiles from the single-z-plane image along the dashed line are shown at the bottom. **b** Snapshots from imaging of EGFP-Rpb3 and mCherry-Rpb1 in an embryo injected with 2% formaldehyde as described above. Intensity profiles from the single-plane image along the dashed line are shown at the bottom. All scale bars, 1 μm. **c** Violin plot with overlaid box plot showing the distribution of the x-y distances between centroids among co-clusters in fixed embryos as described in (**a**, **b**). The central lines of the box plots represent median. Whiskers extend to 1.5 times interquartile range from the box. Data points for outliers are shown. The number of co-clusters (*n*) analyzed using 6 single-z-plane images from 3 different embryos is indicated. *P*-value was calculated by two-sided Mann-Whitney U test. Source data are provided as a Source Data file.

cluster are Zld targets that lack other combinatorial inputs required for transcriptional activation—more or less, the discarded events in a selective maturation process. Alternatively, these Zld-dependent dBrd4 clusters might serve a purpose unrelated to transcription[53].

The apparent direct recruitment of RNAPII to some dBrd4 clusters led us to expect that the amount of RNAPII recruited would correlate with the amount of dBrd4 present. To test this and obtain an objective measure of the colocalization, we assessed the amount of dBrd4 associated with forming RNAPII clusters. During the first 50 s after the detection of RNAPII cluster in cycle 11 (around 2–3 min post mitosis), we observed a moderate positive correlation between dBrd4 and RNAPII intensity (Fig. 4e), consistent with the coupled growth of co-clusters in the initial phase.

Once the co-clusters formed and matured, the subsequent behaviors varied but all culminated in the loss or separation of dBrd4 from the RNAPII clusters. In the example in Fig. 4c (yellow box), the dBrd4 cluster began diminishing in intensity, at about the time that RNAPII reached its peak intensity (Fig. 4c, d, 60 s). This decrease in dBrd4 intensity was followed by a partial disruption of cluster morphology for both dBrd4 and RNAPII. The diminishment or separation of dBrd4 from RNAPII clusters can be observed independently at other sites in this sequence (e.g., asterisk in Fig. 4c). As an objective measure of this reduced association of dBrd4 with RNAPII clusters, we repeated the correlation analysis of dBrd4 and RNAPII intensities at later stages. In contrast to the finding at earlier times (Fig. 4e), we found little evidence for a correlation between 60 and 100 s after the detection of RNAPII cluster in nc11 (Fig. 4f).

The separation of dBrd4 from matured RNAPII clusters could be in part due to movement during imaging of the two channels (less than 1 s). To avoid mobility, we injected 2% formaldehyde into embryos at about 3 min post mitosis and incubated them for another 8 min. The injection of formaldehyde rapidly arrested the progression of nuclear

division cycles, while the sfGFP-dBrd4 and mCherry-Rpb1 clusters were immobilized and retained their fluorescence (Supplementary Fig. 6). In these fixed nuclei, we observed a similar separation of dBrd4 and Rpb1 signals (Fig. 5a). As a control, the same protocol applied to mCherry-Rpb1 and EGFP-Rpb3 confirmed that the two subunits of RNAPII largely overlapped at the foci (Fig. 5b). We quantified the degree of spatial separation by measuring the distance between centroids of overlapping clusters. As shown in Fig. 5c, the average distance between co-clusters of dBrd4-Rpb1 is larger than that of Rpb3-Rpb1 in single-z-plane images.

We conclude that RNAPII clusters form in association with a subset of dBrd4 clusters, but on the longer term, dBrd4 is not retained as a stable constituent in the RNAPII clusters.

## A sustained period of transcription disperses dBrd4 and RNAPII clusters

The above data along with previous studies support a model in which the stepwise modifications of transcriptional hubs ultimately produce locally concentrated pools of RNAPII to stimulate transcriptional initiation. The disruption of both dBrd4 and RNAPII clusters shortly after their maturation led us to ask whether transcriptional engagement of associated genes affects the progressive transformation of transcriptional hubs.

We used a pharmacological inhibitor of RNAPII, α-amanitin, to inhibit translocation and RNA synthesis. We injected α-amanitin before mitosis 11 and performed live microscopy in cycle 12 (Fig. 6a). Inhibition of transcription by α-amanitin slightly enhanced Zld clustering (Fig. 6b, c). More dramatically, in the presence of α-amanitin, dBrd4 clusters first emerged normally in early interphase 12 but then continued to grow without being dispersed, resulting in much more intense clusters in late interphase (Fig. 6d, e). In contrast, this injection of α-amanitin in a prior cycle significantly delayed and reduced RNAPII

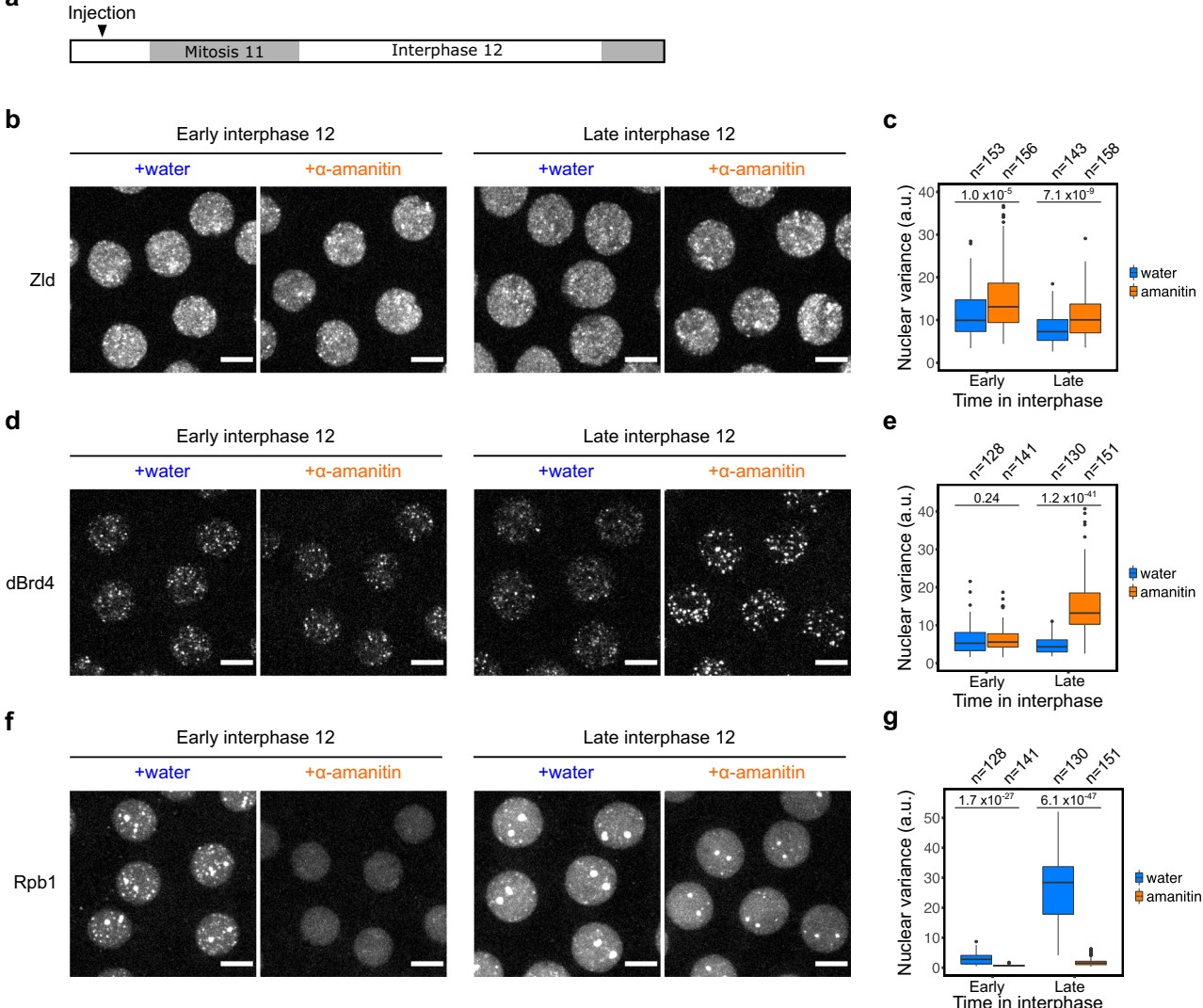

**Fig. 6 | Transcription disperses Zld and dBrd4 clusters but promotes the initial formation of RNAPII clusters. a** The experimental approach to determining the impact of active transcription on the transformation of transcriptional hubs. Either water or α-amanitin was injected into embryos before mitosis 11, and then live imaging was performed during the following interphase 12. **b, d, f** Snapshots from live imaging of mNeonGreen-Zld, sfGFP-dBrd4, or mCherry-Rpb1 in treated embryos as described in (**a**). Maximal projections from the 3 min (early) or 6 min (late) time points in cycle 12 are shown. sfGFP-dBrd4 and mCherry-Rpb1 were recorded together using embryos co-expressing the two reporters. Similar outcomes were observed in 6 embryos for each treatment. Scale bars, 5 µm. **c, e, g** Box plots showing the variance of fluorescent intensities for tagged Zld, dBrd4, or Rpb1 in nuclei from embryos as described above at indicated time points. The central lines of the box plots represent median. Whiskers extend to 1.5 times interquartile range from the box. Data points for outliers are shown. The number of nuclei (*n*) pooled from 6 embryos is indicated above the plot. *P*-values were calculated by two-sided Mann-Whitney U test. a.u., arbitrary unit. Source data are provided as a Source Data file.

clustering (Fig. 6f, g). These results indicate that transcription is dispensable for Zld and dBrd4 clustering but suppresses their persistence either by promoting their decay or suppressing new cluster formation. Since transcription is also needed for rapid recruitment of RNAPII, the stabilization of Zld and dBrd4 clusters by α-amanitin might reflect a direct contribution of transcription and/or a role of RNAPII in the destabilization. In any case, the effects of transcriptional inhibition indicate that transcription normally makes two contributions to the maturation of transcriptional hubs, mildly enhancing the dispersal of Zld and more significantly dBrd4, as well as promoting the accumulation of RNAPII.

To more specifically dissect the effects of transcription on the dispersal of transcriptional hubs, we injected α-amanitin right after the emergence of RNAPII clusters at about 3 min in interphase 13 (Fig. 7a). dBrd4 clusters dispersed within 5 min in the control embryos but persisted in the α-amanitin-injected embryos (Fig. 7b, c). Furthermore,

the non-HLB RNAPII clusters also dispersed within 5 min in the control embryos but persisted in the α-amanitin-injected embryos (Fig. 7d, e). To test whether this stabilization of RNAPII clusters occurs in association with transcribed genes, we performed similar experiments in embryos expressing EGFP-Rpb3 and MCP-mCherry with a *hbP2-MS2* reporter[10]. The *MS2* cassette is inserted in the 5′UTR of the reporter, allowing us to visualize nascent transcripts that have recently initiated. Previous studies showed that most of the MCP foci for this *hbP2-MS2* reporter are found in transient association with RNAPII clusters[12]. Indeed, in control embryos injected with water, the MCP foci colocalized with RNAPII clusters transiently in the beginning of the time course, and as MCP foci grew and expanded, RNAPII clusters were dispersed (Fig. 7f and Supplementary Fig. 7). The late injection of α-amanitin arrested the growth of MCP foci (Fig. 7g, right). Since α-amanitin blocks elongation of transcripts, we interpret the persisting MCP signal as preexisting nascent transcripts "frozen in their tracks",

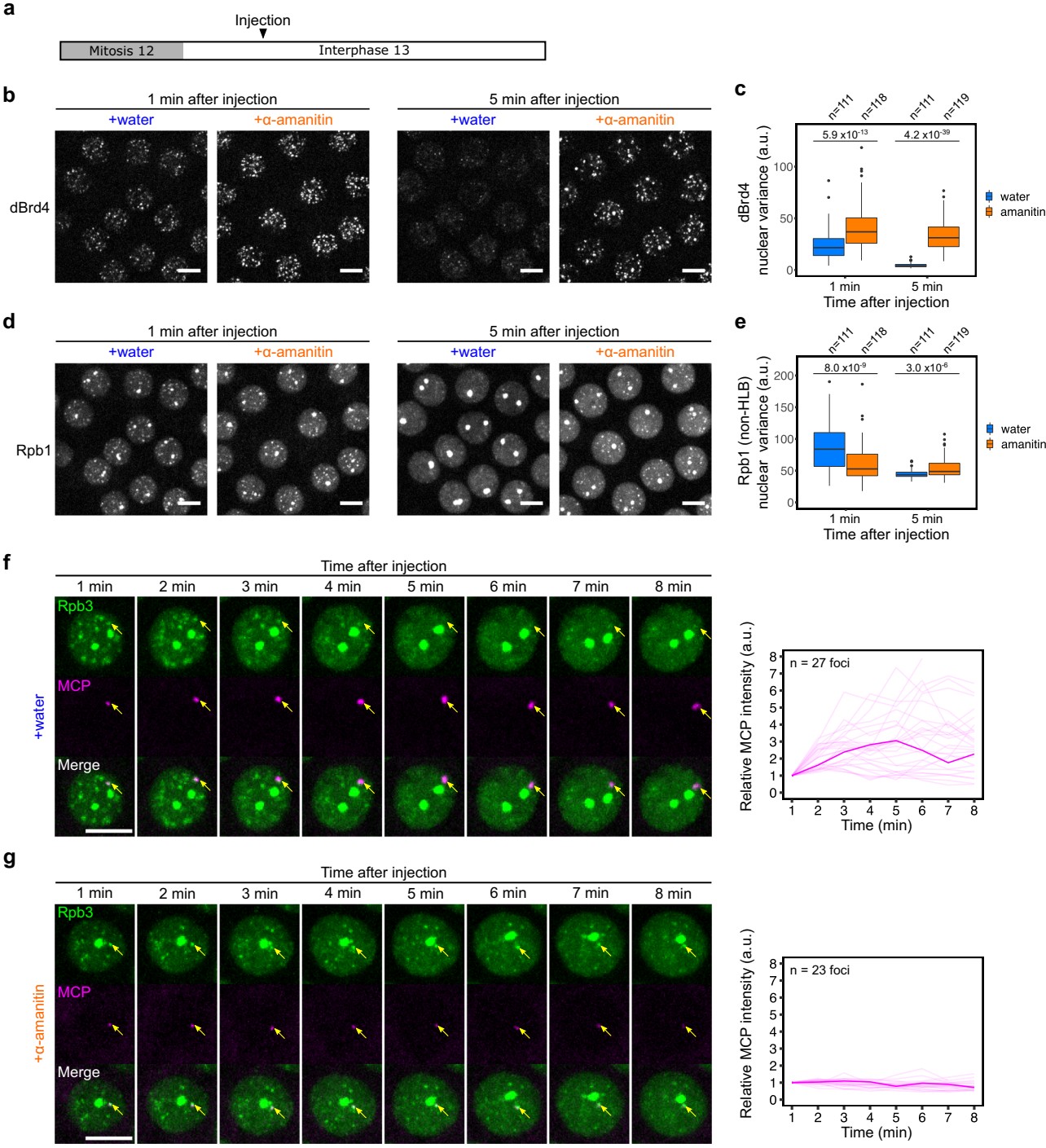

**Fig. 7 | A sustained period of transcription disperses dBrd4 and RNAPII clusters. a** The experimental approach to determining how ongoing transcription impacts established transcriptional hubs. Either water or α-amanitin was injected into embryos 5 min after anaphase-onset of mitosis 12, at which point embryos were about 3.5 min into interphase 13. Live imaging was then resumed 1 min after injection. **b**, **d** Snapshots from live imaging of sfGFP-dBrd4 and mCherry-Rpb1 at indicated time points in embryos as described in (**a**). Maximal projections are shown. Similar outcomes were observed in 3 embryos for each treatment. Scale bars, 5 μm. **c**, **e** Box plots showing the variance of fluorescent intensities for sfGFP-dBrd4 or mCherry-Rpb1 (excluding areas at HLBs) in nuclei from embryos as described above at indicated time points. The central lines of the box plots represent median. Whiskers extend to 1.5 times interquartile range from the box.

Data points for outliers are shown. The number of nuclei (*n*) pooled from 3 embryos is indicated above the plot. *P*-values were calculated by two-sided Mann-Whitney U test. **f**, **g** (left), Representative stills from live imaging of EGFP-Rpb3 and MCP-mCherry in embryos carrying the *hbP2-MS2* reporter and injected with water (**f**), or α-amanitin (**g**), as described in (**a**). The yellow arrows point to positions of the MCP foci. Maximal projections are shown. Scale bars, 5 μm. **f**, **g** (right), Changes in MCP foci intensity relative to the 1 min time point after injection. Only those MCP foci that emerged before the movies started were included in the analysis. Data from multiple nuclei of a single embryo are shown, with the accentuated line corresponding to the MCP focus in the represented stills. Similar outcomes were observed in 5 embryos. a.u., arbitrary unit. Source data are provided as a Source Data file.

an interpretation that is supported by the more complete suppression of MCP signal in nuclei without pre-existing foci prior to injection (Supplementary Fig. 7). Notably, the RNAPII clusters, including those initially associated with MCP foci, persisted in the α-amanitin-injected embryos without dispersing (Fig. 7g and Supplementary Fig. 7). We conclude that an early period of initial transcription is sufficient to nucleate RNAPII clusters, but that sustained transcriptional activity is required for the normal dispersal of both dBrd4 and RNAPII clusters.

## Discussion

It has long been recognized that the compartmentalization of transcriptional machinery is a fundamental aspect of eukaryotic gene control. Early cytological studies revealed discrete clusters of RNAPII and nascent transcripts, which were speculated to be stable "transcription factories"[54]. Subsequent studies show that rather than genes being recruited to stable factories, numerous factors form hubs or liquid-like condensates transiently at active genes. This leaves open the questions of what governs the dynamics of transcriptional hubs/condensates and how their emergence and dispersal are linked to transcript synthesis. In this study, we used real-time approaches to dissect upstream events in transcriptional initiation whose timing is constrained and synchronized in early *Drosophila* embryos by coupling to the rapid cell cycles. We document a cascade of dependencies paralleled by a temporal cascade of cluster formation. Our findings indicate that transcriptional hubs directionally pass through a series of intermediate states with different composition, rather than simply enriching all the factors involved in initiating transcription. Specifically, the pioneer TF Zelda acts through coactivators dCBP and dBrd4 to indirectly concentrate pools of RNAPII near promoters. We also show that inhibition of transcription by α-amanitin stabilizes dBdr4 and RNAPII clusters, indicating that transcription directly or indirectly promotes dispersal of transcriptional hub components resulting in negative feedback. We suggest that the progressive maturation of transcriptional hubs coupled with a negative feedback-loop stimulates a rapid but self-limiting burst of transcription in the early rapid embryonic cycles (Fig. 8). Our findings have striking parallels to the proposal that non-equilibrium dynamics of transcriptional condensates make direct contributions to sequential transcriptional bursts in the longer cell cycles of more mature cells[55,56].

The dynamic nature of transcriptional hubs we described herein is distinct from the well characterized transcriptional condensates at nucleoli or histone locus bodies, which are stable compartments and incorporate multiple functionally related components[5]. The dynamic process with its multiple transitions might serve to add precision and sophistication to transcriptional control. First, transitions between discrete steps could provide proofreading steps that test the stability of intermediate complexes to filter out stochastic noise and increase regulatory specificity. Second, additional regulators might promote or prevent passage through the different transitions, thereby allowing the transcriptional hubs to integrate multiple inputs to generate the intricate spatiotemporal expression of developmental genes. In line with these ideas, our data show that the transitions from Zld clusters to dBrd4 and then to RNAPII are each associated with a decline in the number of clusters, suggesting that the maturation of transcription hubs is selective at successive steps. It will be important to learn how this feature contributes to the extraordinary accuracy with which the graded and combinatorial inputs generate transcriptional outputs.

The molecular mechanisms that drive the sequential transformation of transcriptional hubs remain to be fully determined. During the first step, Zld and dCBP might directly interact with each other or undergo co-condensation[57]. Alternatively, open chromatin established by Zld could facilitate binding of additional TFs that interact with dCBP[34,58,59]. However, it should also be kept in mind that TFs might inhibit deacetylation to indirectly enhance local dCBP-dependent acetylation. In any case, it seems likely, but not yet demonstrated, that dCBP acts by increasing local acetylation to recruit the reader dBrd4. Although dBrd4 might simply bind to histone marks such as H3K27ac, the acetylation of transcription machinery could also be involved in recruiting dBrd4[60]. Upon crossing a concentration threshold, dBrd4 clustering might be promoted by multivalent interactions mediated by its own IDR. While the initial clustering of RNAPII appears to spatially coincide with dBrd4 clusters, the subsequent behavior is not consistent with stable partnership, as dBrd4 is lost from temporarily persisting RNAPII clusters. Imaging the period of loss of dBrd4 revealed accompanying features that varied between clusters: abrupt physical rearrangement of foci, simple gradual loss of dBbr4 from complexes, and apparent de-mixing of previously colocalized signals to form largely separate dBrd4 and RNAPII clusters. These behaviors may represent different manifestations of progressive modifications of the biomolecular condensate that reduce the interactions that previously stabilized co-residency of dBdr4 and RNAPII. Finally, we observed both positive and negative effects of transcriptional elongation on the dynamics of transcriptional hubs. The initial requirement of transcription for RNAPII clustering might involve the upstream roles of

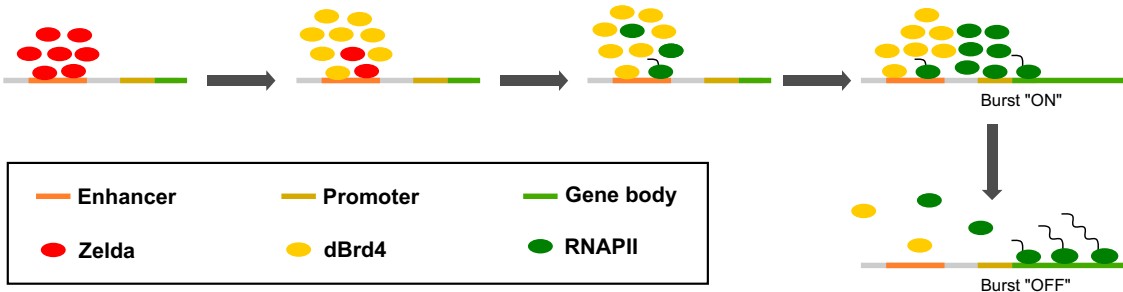

**Fig. 8 | A model for the regulation of transcriptional hubs in early *Drosophila* embryos.** While direct or indirect protein-DNA interactions, schematized as direct contacts, guide the assemblies of protein clusters, the large sizes and the shapes of the clusters suggest further recruitment of proteins via condensate formation or protein-protein interactions. Zelda clusters (red) at the enhancer region act through the lysine acetyltransferase dCBP (not shown) to nucleate dBrd4 clusters (yellow), which then nucleate RNAPII clusters (green) first at the enhancer. We suggest that the RNAPII inefficiently engages DNA to produce enhancer-associated transcripts at this stage. A physical rearrangement then leads to the segregation of dBrd4 and RNAPII and the transfer of RNAPII cluster to the promoter region. Despite some temporal overlap, essential factors recruited at early stages of cluster formation do not persist throughout the maturation of transcriptional hubs. Lack of these factors in the later RNAPII clusters indicates that the initial assembly process does not operate continuously. Locally enriched free RNAPII stimulates a transcriptional burst. Eventually, a sustained period of transcription mediates feedback to destabilize transcriptional hubs, leading to the attenuation of the burst. Note that we anticipate that other steps and numerous other factors, including well recognized transcription factors not included in this simple schematic, will contribute to a sophisticated pathway leading to gene activity: we argue that defining the steps of this pathway will be key in advancing our understanding of the involved mechanisms, and that establishing temporal order and sequential dependencies will guide these advances.

enhancer RNAs in nucleating RNAPII clusters[56]. In contrast, a later sustained period of transcription of the gene body appears to mediate negative feedback to disperse dBrd4 and RNAPII clusters. This could be explained by a suggested disruption of multivalent interaction between IDRs by the negative charge of nascent RNA[56], but numerous other less direct mechanisms might be responsible. While our results reveal the timing and coordination of upstream events required for transcription, much more work is needed to provide a mechanistic understanding of the observed processes.

Regardless of the molecular details, we expect that similar regulatory principles are employed by evolutionarily diverse transcription factors to mediate transcriptional activation. For example, in the zebrafish embryo, the pioneer factors Nanog, Pou5f3, and Sox19b similarly recruit CBP/p300 and Brd4 to establish transcriptional competence during early zygotic gene expression[61,62]. Activation by estrogen receptor α (ERα) also involves histone acetylation and subsequent recruitment of Brd4[63]. Notably, elegant work has shown that dozens of factors are recruited to the ERα target promoter in a cyclical and sequential fashion[64]. We envision that many of these factors are dynamically recruited to the hubs, and that the enzymatic reactions they carry out contribute to the speed and irreversibility of the transformation of transcriptional hubs. Lastly, we suggest that the formation of transcriptional hubs in early embryos ensures the rapid initiation of a transcriptional burst within a short interphase window; in other biological contexts, the hubs might serve additional functions such as bridging enhancers and promoters or coordinating expression of multiple loci[65–67]. The *Drosophila* embryos will provide a powerful system to dissect the relationship between transcriptional hubs, chromatin interactions, and transcription dynamics.

## Methods

### Fly stocks
*Drosophila melanogaster* were maintained on standard cornmeal-yeast medium at 25 °C. Flies were transferred to egg-laying cages 2–3 days before experiments, and embryos were collected on grape juice agar plates with yeast paste. Fly lines used in this study are listed in Supplementary Table 1. Homozygous or heterozygous females used for embryo collection and experiments are listed in Supplementary Table 2.

### Embryo mounting for live imaging
Embryos were collected in egg-laying cages, dechorionated with 40% bleach, washed three times with water, and then transferred onto grape juice agar plates. Embryos were aligned and transferred to coverslips with glue derived from double-sided tape using heptane. The embryos were then covered with halocarbon oil (1:1 mixture of halocarbon oil 27 and 700) for microscopy. The preparation of embryos and microscopy were performed at room temperature ( ~ 22 °C).

### Microinjection
Once aligned and glued to a coverslip, embryos were slightly dehydrated in a desiccation chamber for 7–9 min before being covered in halocarbon oil and subjected to microinjection. JabbaTrap mRNA was synthesized as described in ref. 31 and injected at 750 ng/μl. α-amanitin (Sigma-Aldrich, #A2263) was dissolved in water and injected at 0.5–1 mg/ml. JQ1 (abcam, #146612) supplied as 10 mM solution in DMSO was diluted fresh with water to 0.1 mM before injection. Formaldehyde from a 16% stock solution (Thermo Fisher, #28906) was diluted fresh with water to 2% before injection.

### Fly crosses for germline RNAi experiments
Since it is the maternal genotype that governs the development of syncytial embryos, we describe the crosses producing the parents of the experimental embryos. The following P0 crosses were set

up to generate females for embryo collection in germline RNAi experiments.

1. For visualizing EGFP-Rpb3 following knockdown of a gene targeted by a specific shRNA:
   *EGFP-Rpb3; UASp-shRNA* (females) x *EGFP-Rpb3, Mat-tub-Gal4; Mat-tub-Gal4* (males)
2. For visualizing mNeonGreen-Zld following knockdown of a gene by a specific shRNA:
   *UASp-shRNA* (females) x *mNeonGreen-Zld; Mat-tub-Gal4* (males)
3. For visualizing sfGFP-dBrd4 following knockdown of a gene targeted by a specific shRNA:
   *sfGFP-dBrd4;; UASp-shRNA* (females) x *sfGFP-dBrd4;; Mat-tub-Gal4* (males)

The resulting F1 progeny were either homozygous (1 and 3) or heterozygous (2) for the tagged locus to be imaged and expressed the shRNA of interest in the female germline under the influence of *Mat-tub-Gal4*. The progeny were crossed with their siblings and transferred to egg-laying cages for collection of the experimentally imaged embryos (see also Supplementary Table 2). To increase the Gal4/UASp induction of shRNA expression in the egg laying females, F1 progeny were grown at 27 °C from larval stage onward.

### CRISPR-Cas9 genome editing
DNA oligos for sequences of sgRNA targeting the 5' end of the dBrd4-encoding *Fs(1)h* locus (5'-CGGTGGCTCACTGGACGACA-3') were ordered from Integrated DNA Technologies (IA), annealed and cloned into the expression vector pU6-BbsI-chiRNA using standard protocols[68]. To make donor plasmids, about 1 kb of homology arms upstream and downstream of the start codon were amplified from the genomic DNA of the *nos-Cas9* (III) flies. HaloTag and sfGFP DNA fragments with 5x Gly-Gly-Ser linker were amplified from plasmids previously made in the lab by PCR. Vector backbone was amplified from pDsRed-attP with the 3xP3-DsRed cassette removed. All DNA fragments were then gel-purified and assembled by Gibson assembly. The sgRNA-expressing and donor plasmids were sent to Rainbow Transgenic Flies for microinjection. After injection, surviving adults were crossed to *yw, N/FM7c* balancer flies and screened by PCR for successful knock-in. Transformants were backcrossed with wild type (Canton S: *w$^{1118}$*) at least three times before performing experiments.

### Labeling of HaloTag-dBrd4 in embryos
We noticed that the labeling of HaloTag-dBrd4 by fluorophore-conjugated ligand frequently caused mitotic defects, which could be alleviated in embryos from females heterozygous with untagged *dBrd4* allele. We thus used embryos from the *HaloTag-dBrd4/+* females for experiments. HaloTag TMR ligand was dissolved in DMSO at 5 mM as stock solution and diluted fresh to 10–15 μM with water for microinjection. Injected embryos were incubated for at least 10 min at room temperature before imaging.

### Molecular cloning and phiC31-mediated transgenesis
To construct the donor plasmid for *UASp-JabbaTrap-bcd3'UTR*, 834 bp of *bicoid* 3'UTR was amplified from the genomic DNA of wild-type flies and inserted into *pUASp-attB-JabbaTrap* plasmid backbone by Gibson assembly. The plasmid was then injected into *attP112 (III)* lines for phiC31-mediated integration by BestGene.

### Fly crosses for JabbaTrap experiments
The following P0 cross was set up to obtain progeny used for the JabbaTrap dBrd4 cage.

*sfGFP-dBrd4, mCherry-Rpb1;; UASp-JabbaTrap-bcd-3'UTR* (females) x *sfGFP-dBrd4, mCherry-Rpb1; Mat-tub-Gal4* (males)

The following P0 cross was set up to obtain progeny used for the JabbaTrap control cage.

*mCherry-Rpb1;; UASp-JabbaTrap-bcd-3′UTR* (females) x
*mCherry-Rpb1;; Mat-tub-Gal4* (males)

The resulting F1 progeny were grown and kept at 25 °C for experiments, as growing at 27 °C still led to complete female sterility in JabbaTrap dBrd4 embryos. F1 progeny were allowed to mate with their siblings and transferred to egg-laying cages for experiments (see also Supplementary Table 2).

## Embryo fixation and immunostaining

Dechorionated embryos were transferred into a 1.5 ml tube with 500 μl heptane, and then 500 μl of freshly prepared 4% formaldehyde in PBS was added. The tube was vigorously shaken for 20 min at room temperature. After removing the lower aqueous layer carefully and as much as possible, 500 μl methanol was added, and the mixture was shaken for 1–2 min. Devitellinized embryos which sank to the bottom were kept and washed with methanol for three times. Fixed embryos were stored at −20 °C in methanol until use. To perform immunostaining, fixed embryos were rehydrated with 500 μl PBST (0.3% Tween-20) for 5 min at room temperature four times. Embryos were blocked in PBST with 3% donkey serum (Sigma-Aldrich) for 30 min. 1:500 rabbit α-H3K27ac (Active Motif, #39133) was then added and incubated at 4 °C overnight. Next, embryos were washed with PBST for 15 min three times, incubated with 1:500 anti-rabbit Alexa Fluor 488 (Thermo Fisher, #A11008) at room temperature for 1 h, washed with PBST for 15 min three times, and mounted on a glass slide in Fluoromount (Sigma-Aldrich, #F4680). Hoechst 33258 (Thermo Fisher, #H3569) was added at 1:2000 during the second wash after secondary antibody incubation.

## Embryo RNA extraction and RT-qPCR

For each biological replicate, about 100 dechorionated embryos were homogenized by a plastic pestle in TRI Reagent (Zymo Research, #R2050-1-50), and the total RNA was extracted using Direct-zol RNA microprep kit (Zymo Research, #R2060). cDNA was synthesized using Promega GoScript Oligo(dT) (#A2790) following manufacturer's protocol. qPCR mixture was prepared using Bio-Rad SsoAdvanced Universal SYBR Green Supermix (#172-5270) and analyzed on Bio-Rad CFX Connect system. *RpL32* was used as a reference gene, and *dCBP* level was quantified by the ΔΔCt method.

## Western blot analysis

About 100 dechorionated *sfGFP-dBrd4* embryos between 0.5 and 1.5 h old were transferred to 50 μl of RIPA buffer supplemented with protease inhibitors (Pierce, #A32955). Embryos were homogenized on ice by a plastic pestle, and then 50 ul of 2X SDS-PAGE sample buffer was added. After boiling for 5 min, 10 ul samples were separated on 7.5% SDS-PAGE at room temperature and then transferred to PVDF membrane in a cold room. The membrane was blocked with TBST (0.1% tween 20) and 5% milk, blotted with 1:5000 rabbit anti-GFP (Abcam, #ab290) in a cold room overnight, followed by 1:20000 anti-rabbit HRP (Thermo Fisher, #A16096) at room temperature for 1 h. The membrane was incubated with SuperSignal West Pico PLUS Chemiluminescent Substrate (Thermo Fisher, #34577) and exposed to film (Thermo Fisher, #34091).

## Spinning disk confocal microscopy

Imaging was performed on an Olympus IX70 microscope equipped with PerkinElmer Ultraview Vox confocal system. Movies following the mid-blastula transition in nc14 were acquired with a 60x/1.40 oil objective; all other movies were acquired using a 100x/1.40 oil objective. With the 100x objective, a field of 67 × 67 μm was recorded, which contains about 20–30 nuclei in cycle 12 embryos and about 40–60 nuclei in cycle 13 embryos. Data were acquired using Volocity 6 software (Quorum Technologies). Pixel binning was set to 2 × 2 for imaging of live embryos and 1 × 1 for fixed samples. Focal planes with a 0.5–0.75 μm z-step were recorded at each timepoint. In dual-color imaging, sequential acquisition was performed through channels first then z-planes. Fluorophores were excited with 488 and 561 nm laser lines. Appropriate emission filters were used in most cases. When imaging sfGFP-dBrd4 and mCherry-Rpb1 at higher frame rate (10 s), the "fast sequential" mode without applying emission filters was used, and care was taken to make sure that there was negligible bleed through under the acquisition setting. Images in the same set of experiments were acquired using the same configuration, and laser power was calibrated using a laser power meter (Thorlabs) before each imaging session.

## Image analysis

Data obtained in Volocity were exported as image stacks and background-subtracted with a rolling-ball radius between 50 and 75 pixels in FIJI/ImageJ. All image processing, segmentation, and quantification were performed in FIJI/ImageJ (v2.9.0) and Python 3, using standard and open-sourced libraries such as NumPy and scikit-image[69].

**Quantification of nuclear variance of fluorescent intensity.** We used the variance of fluorescent intensities in the nucleus to estimate the degree of clustering. Binary masks corresponding to nuclei were generated by Otsu's thresholding after Gaussian blurring. When comparing the same embryo over time (e.g., Figs. 3b, 4b), variance in the combined nuclear mask was measured at each time point. When comparing embryos with different treatments, variance in single nuclei was measured, and nuclei from multiple embryos were pooled together for analysis. When quantifying variance of non-HLB RNAPII, the masks for HLBs were generated by manual thresholding and excluded for analysis.

Note that this method will also detect background noise in addition to real clustering; nonetheless, the heterogeneity contributed by real clusters is much larger than background noise. We thus used this method to objectively compare changes in clustering over time or between different groups, instead of trying to perform segmentation to identify and quantify cluster intensity.

**Manual tracking and quantifications of dBrd4 and Rpb1 clusters.** Time-lapse dual-color imaging of sfGFP-dBrd4 and mCherr-Rpb1 was performed at a frame rate of 10 s. We tracked the co-clusters manually based on local nearest neighbors and the dynamics of fluorescent intensity. The integrated intensity was then measured in a small circle, and the data were normalized between 0 and 1 range.

**Correlation of Rpb1 and dBrd4 intensity at RNAPII clusters.** mCherry-Rpb1 clusters were detected using the LoG (Laplacian of Gaussian) detector in the TrackMate tool in FIJI[70], with an object diameter of 4 pixels (0.5 μm). The resulting objects were further filtered manually by Rpb1 intensity to remove noise and the large HLB clusters. Object labels were exported from TrackMate as TIFF and used for quantification of fluorescent intensity in Python. For each identified object, the integrated intensities for mCherry-Rpb1 and sfGFP-dBrd4 were calculated after subtracting a nucleoplasmic background, which is determined as the median of nuclear intensity. Pearson's correlation between the integrated intensities of Rpb1 and dBrd4 was performed in R.

**Quantification of spatial separation between co-clusters.** For each pair of dual-color imaging (dBrd4-Rpb1 and Rpb3-Rpb1), a total of 6 single-z-plane images from 3 fixed embryos were used for analysis. Segmentation of clusters was performed using the Trainable Weka Segmentation tool in FIJI[71] and then filtered by a minimal size of 2 pixels. Co-clusters were identified by the spatial overlap of at least 1 pixel. The distance between centroids of co-clusters was then measured.

**Tracking of MCP foci and quantification of fluorescent intensity.**
MCP foci were detected, tracked, and quantified by TrackMate[70]. The
LoG detector and an object diameter of 8 pixels (1 μm) were used. Data
for each track were individually normalized to the initial intensity in
the first frame. MCP foci that emerged after the first frame were
excluded from analysis.

## Statistics and reproducibility

All experiments were repeated at least three times with similar out-
comes, and representative images are shown in the figures. Two-tailed
Mann-Whitney U tests were performed in R (v.3.2.1). A value of $P < 0.05$
was considered statistically significant. Additional details can be found
in the corresponding figure legends.

## Reporting summary

Further information on research design is available in the Nature
Portfolio Reporting Summary linked to this article.

## Data availability

Raw imaging data and original Python scripts generated in this study
have been deposited at the Zenodo database (https://doi.org/10.5281/
zenodo.8136965). Any additional information is available from the
corresponding author upon request. Source data are provided with
this paper.

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

## Acknowledgements

Stocks obtained from the Bloomington Drosophila Stock Center (NIH P40OD018537) were used in this study. We are grateful to Christine Rushlow for providing the *UASp-zld.shmr* fly line and to Michael Eisen for sharing the *mNeonGreen-Zld* fly line. We thank Melissa Harrison for discussing their parallel work on *nejire*. This work is supported by National Institutes of Health grant R35GM136324 (to P.H.O).

## Author contributions

C.-Y.C.—Conceptualization, Methodology, Formal analysis, Investigation, Writing-Original Draft, Visualization. P.O'F—Conceptualization, Writing-Original Draft, Supervision, Funding acquisition.

## Competing interests

The authors declare no competing interests.
