## [Peer Review File · Nature Communications]

Stepwise modifications of transcriptional hubs link pioneer factor activity to a burst of transcriptionREVIEWER COMMENTS

Reviewer #1 (Remarks to the Author):

The manuscript by Cho and O'Farrell analyses the clustering dynamics of the Zelda pioneer transcription factor with the co-activator dBrd4 and RNA polymerase II using live imaging of fluorescently tagged endogenous proteins in early developing drosophila embryos. Using this approach, the authors can decipher the chronology of interactions of these various proteins within observable clusters. In parallel, they aim to decipher the causal requirements between these proteins for the formation of these clusters using genetic perturbations, i.e. RNAi or cytoplasmic sequestrations of those nuclear proteins induced by direct injections in embryos of Jabba Trap mRNA for which they have a long-lasting expertise. They propose that this chronology is reflecting the dynamics of interactions occurring at transcriptionally activated loci. Using an MS2 reporter for the hunchback gene promoter (P-element insertion), they show that this transcribing locus coincides with a PolIII cluster. As they observe that the MS2 signal persists after the disappearance of the PolIII cluster, the authors propose that transcriptional elongation might disperse those protein clusters and have thus a negative feed-back on transcriptional bursting.

In general, the work presented in this manuscript is interesting and nicely call for the power of live cell imaging to decipher the dynamics of the transcription activation process. The experimental representative data shown are quite convincing but their quantitative and statistical analyses need to be strengthened. Also, concerning the use of the MS2 reporter, the authors should be more careful with their conclusion as the observation of the MS2 signal at a given time is not necessarily an indication that the promoter is transcribing at this time point (particularly if the MS2 cassette is at the 5' end of the transcribed sequence). Finally, the authors must be a bit careful in their phrasing because the link between the co-clusters that they observe, and the transcription process itself assumes that a PolIII cluster is transcribing, and this link is therefore indirect.

Specific comments :

1 - As mentioned in the text (line 101), the fluorescent clusters observed with the various tagged proteins are numerous and of various sizes. Concerning the statistical analysis, only the number of experiments performed and/or the number of embryos analyzed are mentioned. It is however important to know how many clusters were used for the curves presented in Fig 2b , Fig 2d, Fig 4b, Fig 5a and Fig 5b (the 3 last ones have no error bars) and if any thresholding was used to choose the clusters in the analyses. Also, the number of nuclei analyzed per embryo is important to mention.

In the same line, when looking for co-clustering, some dBrd4 clusters never co-cluster with RNAPII clusters and it is mentioned in the text (line 216) that this suggests additional inputs. If this conclusion is one possibility, there are many others : for instance, it is possible that some dBrd4 clusters are not linked to transcription but to another nuclear process involving histone acetylation. Also, given these differences among clusters, it is a bit surprising that the authors used only the total intensity of the

clusters (without distinguishing them) in their analysis. It will be useful to know how many of co-clusters are observed for each combination of proteins and if there is a correlation between the size of the clusters, their intensity and their ability to co-cluster.

Finally, the quantification of cluster intensity is required in Fig 1, Fig 6 and Fig Supp 7 because it is very difficult to judge. Also, the quantification of the mNeonGreen-Zld would help very much as the color code might not clearly show differences (why not using black and white instead ?).

2 – Concerning the genetics, the dBrd4- context correspond to the Jabba trap cytoplasmic sequestering using the Bcd-3'UTR to express the sequestering Jabba only after fertilization. As the Bcd3'UTR also anchored the RNA at the anterior pole, it is likely that the Jappa Trap mRNA will be anchored at the anterior pole and therefore that the dBrd4 sequestering will be very efficient around the anterior pole and less efficient in the middle of the embryo. It will be important to show it and to quantify the staining of Rbp1 shown in Figure 3d. It is also possible that the version of the Bcd-3'UTR used only control expression after fertilization but not the anchoring of the RNA at the anterior pole, but if this is the case, it has to be mentioned in the manuscript.

Also, it is very difficult to follow the exact genotype of the embryos analyzed. These needs to be indicated in the Figure legend or in the Material & Methods or supplementary Table 1. For instance, in Figure 1a, the embryos injected were likely “w, mCherry-Rbp1, sfGFP-Zld” and in Figure 1c, the embryos were likely from a cross between females Mat-tub-Gal4 with males carrying the shRNA concerned but it is not clear how many copies of the EGFP-Rbp3 transgene they carry.

3 – The authors observe the spatial separation of the dBrd4 and RNAPII clusters (Figure 5). This observation is interesting even though a bit a surprising because in principle, the distance between the two complexes at the expected activated loci is supposed to be below the resolution limit of the imaging. Nevertheless, the quantification shown in Figure 5a and 5b are very appealing. As mentioned already above, quantifications on many clusters are necessary to support the claims. This might also include to work on the temporal issue to determine if the spatial separation is observed during the whole period when the co-clustering occurs. Also, since the authors have used the MS2 reporter, it is important to determine if a similar observation is made between dBrd4 and the transcribing loci (MS2) and between PolIII and the transcribing loci (MS2).

4 – To detect a transcribing locus, the authors use a MS2 reporter for the hunchback gene which carries the MS2 cassette in 5' of the transcribed sequence. This position of the MS2 cassette allows to detect the transcribing RNAPII as soon as they elongate, and the fluorescent signal accumulates very easily and very rapidly at the transcribed locus. This signal is however so strong that the dynamic range of the fluctuation does not allow to catch easily the OFF-time of the promoter. It is therefore not very surprising that the locus is still showing MS2 signal while it is not anymore in an RNAPII cluster. This must be explained properly in the text.

Minor points :

1 - The authors should indicate somewhere for each of the stills if they correspond to maximal z-projections (it is only mentioned in Figure 2a legend) or if they are just one z-stack.

2 – It is not clear why the curves shown in Fig 2b, Fig 2d, Fig 4b, Supp Fig 4 are only starting at 20 or 30 seconds while the signals are above 0.

3 – The sentence line 136 – line 137 : “Additional evidence....” is not clear and not useful for the rest of the manuscript.

4 – The term “chromatin-bound” (line 161) must be explained. No experiment is really showing that the clusters detected are bound to chromatin.

4 – The transgene allowing the expression of the MCP-mCherry is not referenced in the Supplementary Table 1.

Reviewer #2 (Remarks to the Author):

This study by Cho and O’Farrell uses high-resolution live imaging of tagged proteins to visualize the transcriptional machinery during zygotic genome activation in the early fly embryo. The authors demonstrate that the pioneer transcription factor Zelda, the acetyltransferase dCBP, and the reader BET protein dBRD4 form nuclear clusters / hubs during interphase of early nuclear cycles. By assessing the consequences of knockdown of the proteins, the authors further show that Zelda works with dCBP to nucleate dBRD4 clusters, which are necessary for the formation of RNAPII hubs. This study advances our knowledge of the temporal order of events that lead to transcriptional initiation. I would recommend publication after the following questions and concerns are addressed:

1) It is difficult to tell if Zld and Rpb1 show very little overlap from one z-plane image. In Fig 1a, the three most distinct and brightest Zld clusters (150-210, bottom left) actually do overlap with the three most distinct Rpb1 clusters. Has a quantification been done of the number of clusters for each protein and then how many of those overlap for each time point? In other words, is it possible to put a number on “rarely colocalized?”

2) Line 174. Here the authors make note of the observed overlap of Brd4 and Zld clusters, but from looking at Fig. 2e, there are only a few instances of obvious overlap, similar to Zld and Rbp1. Since there is no rigorous quantification of clusters in either experiment, it is difficult to conclude if there is more overlap between Zld and Brd4 than Zld and Rbp1.

3) Line 79. The authors appear to be hypothesizing that the lack of visual colocalization between Zld and Rbp1 is because the clusters are not only transient, but dependent on a specific temporal order of

events such that you would not expect them to colocalize. It is known from super resolution imaging (Mir and Eisen) that Zelda binds very transiently to its target sequences, so this hypothesis is plausible even if there were no intermediate steps. Was a more precise tracking analysis similar to the cluster tracking done for dBrd4 and Rbp1 (in Fig. 4) done for Zld and Rbp1 to rule out colocalization?

4) Lines 133. Is there evidence that the Zld clusters actually represent Zld bound to its target motifs? Was this shown here or elsewhere? If so, please show or reference. If not, the suggestion “Zld presumably at its target DNA sites” is overstated.

5) Line 251. Where is the data that shows dBrd4 is “redistributed” to the HLBs? How was this measured? At the very least, I do not see a quantification of the signal intensity of the HLBs over time. From simple glance at the spots, they don’t appear to get bigger.

6) Line 88. Does transcriptional bursting occur in early nuclear cycles? Seems nc11-12 are too short for multiple bursts to occur, nevermind be visualized.

7) Fig. 3. Please label frames in figure since in the text you refer to “frames” versus seconds.

Reviewer #3 (Remarks to the Author):

Review: Cho and O’Farrell, ‘Stepwise modifications of transcriptional hubs link pioneer factor activity to a burst of transcription’

In this manuscript, the authors investigate the formation and impact of transcriptional hubs. They address this important question in the early *Drosophila* embryo, amenable to live imaging and genetic manipulations. In particular, they examine the spatio-temporal clustering of major transcriptional regulators such as Pol II (Rbp1 and Rbp3), the acetyltransferase dCBP, the transcriptional co-activator dBrd4 and the pioneer factor Zelda, by utilizing through live imaging from endogenously tagged loci. The key finding of this study is the sequential formation of transcriptional hubs constituted by these factors as well as their cascade of dependencies. Except for HLB loci (which appear to be controlled by an alternative cascade), Zelda clusters seem to come first, working through co-activators like dBrd4 and dCBP to then elicit the formation of Pol II clusters. Timing and dependencies are clearly demonstrated via elegant genetic manipulations (RNAi or Jabba trap + live imaging). The manuscript ends with the surprising finding that transcription feedbacks on a subset of transcriptional hubs, namely dBrd4 and Pol II clusters, in order to promote their dispersal.

In general, I think the findings of this paper are original, well-presented and constitute an important

contribution to the field of gene regulation during development. However, the manuscript lacks quantifications to back up most of the conclusions. While I am convinced by the findings regarding the transcriptional hub dynamics (Fig1 to 5), I am a bit more skeptical about the last part of the manuscript regarding the effect of sustained transcription on clustering (see below point 2).

Major comments:

1. Most of the results are descriptive and not quantified. The authors generally mention in the figure legend that the results were observed in a number x of embryos, but without extracting quantitative information from these embryos.

Since the data have already been obtained and since the procedure for image analysis is already in place (for example Fig 2b), it would be relatively easy to strengthen each figure with the appropriate quantifications.

General number concerning hub kinetics could be extracted from the data to learn:

how fast do hubs form? how fast do they dissolve? how long do they interact with DNA? how long do they interact with a given TF/GTF?

Here are a few examples of findings that would be solidified by quantifications:

-Fig1a, bottom: signal at HLB clusters seems not to be affected by Zelda depletion.

-Fig1b/line 116: RNAP II clusters emerged, but infrequently overlapped with Zelda cluster.

-Fig 2e.f/line 173: visual inspectionnewly formed dBrd4 clusters occasionally colocalized with Zelda clusters.... Maybe quantify % of coloc at different time points.

-Figure 4: temporal disconnect between dBRD4 and RNAPII clusters. The authors could display the distribution of ' Δt ', lag time between peak signal for each cluster and envisage correlation analyses.

-Figure 5: same point as before, for spatial disconnect: quantify distance between clusters to observe the frequency of colocalization.

-Figure 6a/line 239: dBrd4 clusters become more intense. Quantify size and intensities.

-Figure 6b/line 258: MCP TS foci: quantify over time.

2- Conclusions regarding the impact of a sustained transcription on cluster decay are not solidly demonstrated.

-The inhibition of transcription with alpha-amanitin injection is difficult to interpret, as the timing of action of this drug is poorly characterized in the fly embryo. In mammalian cells the delay between drug application and effect can require several hours. The fly embryo is uncellularized but the large volume the drug has to diffuse into remains a challenge. Here, MCP-MS2 dots are still observed minutes after drug injection, indicating transcription did not arrest. Alternatively, the remaining spot corresponds to Pol II blocked with MS2-nascent mRNAs, exhibiting high retention of MCP. One can decipher this by fixing embryos after drug injection and performing smFISH on MS2 in the absence of MCP detector protein or perform FRAP in live MS2/MCP embryos.

-Authors should perturb transcription with alternative methods: genetically (RNAi of overexpression of transcription initiation/elongation factors) or with other drugs (flavopiridol, triptolide, DRB).

-The observations are made at stages pre-ZGA, before the major transcription wave. It would be interesting to analyze how Brd4 and Pol II clusters behave during nc14, and whether they would tend to

be more disperse.

3-Since Zelda, Pol II, dCBP and dBrd4 Chip-seq results are available for early Drosophila embryos, it would be interesting to discuss the temporal cascade discovered in this study with the overlap of bound targets.

More generally, the 'DNA' content of these clusters is not discussed in the manuscript, yet it represents an important piece of the puzzle.

4-Discussion regarding the implication of the sequential formation/dissociation of clusters with respect to transcriptional bursting (line 276). I would nuance the statement, as the authors only monitor a specific timescale of transcriptional dynamics. We now know that bursting is multiscale (Lammers COGD), ranging from seconds to days. Thus TF/GTF/co-activator clustering may impact a specific timescale of transcriptional bursting but cannot be the sole source of bursting.

Minor comments:

-A figure with a model would be appreciated.

-the authors use clusters or hubs. Is there a conceptual difference?

- Why JabbaTrap is used via RNA injection for depleting Zelda and with genetics (UAS-JabbaTrap-bcd3'UTR)?

-The efficiency of depletion of Zelda with Jabba trap is not shown.

Could the author back up the requirement of Zelda to initiate RNAPII clustering with Zelda RNAi approach?

-Line:628 explanations of how the quantification was performed is unclear: what is a boundary? How was it defined? What is the zero in the x axis?

-to be rigorous, the genotype of embryos depleted for dBrd4 or Zelda with the Jabba trap trick should not be named 'dBrd4-' or 'Zelda-'.

-line 131: why shifting to Rbp3 while the rest of the figure employed Rbp1?

-Endogenous tagging of dBrd4 can be difficult (and the authors mention some issues in the methods: line 350). The authors claim that embryos laid by sfGFP/Halo-dBrd4 mothers have a similar hatching rate than controls (line150). Could the authors show the hatching rate quantification in a supplementary figure?

-Figure 3d: in dBrd4-, in nc14, nuclei seem to show only 1 HLB cluster. Is this true? Can the author comment on this result (if true) ?

-missing references:

The mitotic retention of dBrd4 is published in and should be cited (Ringrose lab).

The effect of Zelda on enhancer hub formation is analyzed in Espinola et al and not discussed in the manuscript.

Overview of responses to reviewers

We are grateful that the reviewers recognize the values of our real-time approach in establishing the chronology of events during the onset of transcription in the *Drosophila* embryo. All three reviewers found the manuscript interesting but sought further quantification of the presented images to either bolster or extend the interpretations offered. We have tried to respond to all the specific requests. Before presenting these details, we wanted to point out that, in some cases, the limitations of the data or different perspectives have led us to take different routes to what we hope is the same end of supporting a convincing description of upstream events involved in initiating transcription.

Below please find a point-by-point response to the reviewers' comments. Our responses are highlighted in blue.

--

Comments from Reviewer #1:

The manuscript by Cho and O'Farrell analyses the clustering dynamics of the Zelda pioneer transcription factor with the co-activator dBrd4 and RNA polymerase II using live imaging of fluorescently tagged endogenous proteins in early developing drosophila embryos. Using this approach, the authors can decipher the chronology of interactions of these various proteins within observable clusters. In parallel, they aim to decipher the causal requirements between these proteins for the formation of these clusters using genetic perturbations, i.e. RNAi or cytoplasmic sequestrations of those nuclear proteins induced by direct injections in embryos of Jabba Trap mRNA for which they have a long-lasting expertise. They propose that this chronology is reflecting the dynamics of interactions occurring at transcriptionally activated loci. Using an MS2 reporter for the hunchback gene promoter (P-element insertion), they show that this transcribing locus coincides with a PolIII cluster. As they observe that the MS2 signal persists after the disappearance of the PolIII cluster, the authors propose that transcriptional elongation might disperse those protein clusters and have thus a negative feed-back on transcriptional bursting.

In general, the work presented in this manuscript is interesting and nicely call for the power of live cell imaging to decipher the dynamics of the transcription activation process. The experimental representative data shown are quite convincing but their quantitative and statistical analyses need to be strengthened. Also, concerning the use of the MS2 reporter, the authors should be more careful with their conclusion as the observation of the MS2 signal at a given time is not necessarily an indication that the promoter is transcribing at this time point (particularly if the MS2 cassette is at the 5' end of the transcribed sequence). Finally, the authors must be a bit careful in their phrasing because the link between the co-clusters that they observe, and the transcription process itself assumes that a PolIII cluster is transcribing, and this link is therefore indirect.

We thank the reviewer for the positive views on our manuscript. We have made efforts to address the reviewer's general concerns. We have significantly revised our manuscript to provide additional quantification and statistical analysis, discussed in more detail below. We have adjusted our wording of the relationship between the MS2 signal and the promoter activity. Just to be sure we are on the same footing regarding this relationship, we presume that the reviewer accepts that the emergence of the MS2 signal indicates the past occurrence of transcriptional initiation and some elongation, and that increases in the MS2 signal overtime reflect, with a time lag in the minute-range, ongoing promoter activity. We assume that the reviewer is concerned that the mere presence of MS2 signal, while reflecting persistence of nascent transcripts that are generally thought to be elongating, does not necessarily require ongoing new initiation. We do agree with these concerns and hope that the wording we have used conveys so. Finally, the data detailing the distinctions in the time courses of accumulation of RNAPII and accumulation of an MS2 reporter signal was presented in our previous study (Cho et al., Cell Rep., 2022; PMID: 36261005). In addition, in this previous study, we used a global assay to show that PolIII clusters colocalized with foci of nascent RNA labeled by 5-EUTP in early embryos, and, reciprocally, that the

nascent transcript foci colocalized with PolIII clusters. We have incorporated reference to these previous findings to the Introduction.

Specific comments :

1 - As mentioned in the text (line 101), the fluorescent clusters observed with the various tagged proteins are numerous and of various sizes. Concerning the statistical analysis, only the number of experiments performed and/or the number of embryos analyzed are mentioned. It is however important to know how many clusters were used for the curves presented in Fig 2b , Fig 2d, Fig 4b, Fig 5a and Fig 5b (the 3 last ones have no error bars) and if any thresholding was used to choose the clusters in the analyses. Also, the number of nuclei analyzed per embryo is important to mention.

As the reviewer noted and recognized, the clusters vary substantially in sizes, shapes, and intensity, resulting in several difficulties in image analysis. These technical challenges include distinguishing faint foci from background noise, isolating individual clusters (segmenting), and tracking clusters between frames. We decided to avoid complications and use variance as a surrogate measure of clustering. Clustering into foci increases the heterogeneity in the intensity as compared to a more homogenous distribution without subnuclear localization. Variance of all pixel intensities gives a measure of this heterogeneity and hence clustering. In the initial submission, we used a custom approach with arbitrary thresholding to assess the degree of clustering, reasoning it would better capture the heterogeneity caused by clusters. This original approach was difficult to explain and understand. We have found that simply measuring variance in all pixels works just as well, and now use it as a measure of heterogeneity/clustering. While questions might arise about the quantitative accuracy of variance as a measure of clustering, it is unclear whether any measure can be argued as a numerically accurate quantification when comparing very different distributions, such as those produced by numerous weak Zld clusters and the strikingly localized dBrd4 clusters. Because the meaningful comparisons in the paper involve the changes over time in similar patterns, variance provides an objective and direct measure of differences in clustering. While variance will include a contribution of imaging noise, this is negligible in comparison to the heterogeneity produced by clustering. We have provided a more detailed description of our approach to quantifying clustering in the revised manuscript.

Since the method we used to assess clustering (variance) does not involve enumerating the cluster, we do not have information for the exact number of clusters analyzed using this method. We have, however, provided the number of nuclei pooled from multiple embryos in corresponding figures and figure legends. In the Methods section for confocal microscopy, we also added details of our imaging settings, including the typical number of nuclei recorded per embryo.

In the same line, when looking for co-clustering, some dBrd4 clusters never co-cluster with RNAPII clusters and it is mentioned in the text (line 216) that this suggests additional inputs. If this conclusion is one possibility, there are many others : for instance, it is possible that some dBrd4 clusters are not linked to transcription but to another nuclear process involving histone acetylation. Also, given these differences among clusters, it is a bit surprising that the authors used only the total intensity of the clusters (without distinguishing them) in their analysis. It will be useful to know how many of co-clusters are observed for each combination of proteins and if there is a correlation between the size of the clusters, their intensity and their ability to co-cluster.

We agree with the reviewer that some dBrd4 clusters might not be linked to transcription and thus never recruit RNAPII. We mention this possibility in the revised manuscript. Indeed, the manuscript does describe mitotic clusters of dBrd4, which we think have distinct and important role. On the other hand, clusters that form at a similar time and are similarly dependent on Zld and CBP might be more likely to be related in function.

We also agree that potential differences among clusters are of interest, especially since each is likely to represent a different locus that might be influenced by a unique combination of transcription factors.

However, unless we track individual clusters, we don't even know whether an especially bright cluster in one frame is the same cluster that is especially bright in the next frame. As we have indicated, at least for Zld we cannot reliably track clusters and assess correlation of properties. Because dBrd4 clusters are fewer and more long-lasting, we can track some of them manually, but even here there are uncertainties. By looking in the earlier cycle in cycle 11, where there are fewer dBrd4 clusters at a somewhat faster (but still limited) frame rate, we have been able to track some of the brighter dBrd4 clusters and describe the recruitment of RNAPII to them. We have made efforts to avoid analyses that cannot be rigorously supported by our present data and hope that in future analyses that mark specific genomic loci, we will be able to follow events at recognized and distinct loci.

We argue that despite limitations, this study shows that upstream factors in the pathway do not persist in the more mature clusters—a clear finding that is not consistent with many descriptions of transcriptional activation as recruitment of multiple factors through cooperative interactions to build a stable assembly. We have revised our manuscript, including reordering the figures, to highlight this conclusion. We have left as ambiguous whether Zld promotion of dBrd4 cluster formation involves transient co-clustering, and suggest that such co-clustering might not be required if Zld acts through an intermediary (i.e. histone acetylation) to trigger dBrd4 cluster formation.

As in the initial manuscript, we included data following some individual clusters of dBrd4 to show that some dBrd4 clusters recruit RNAPII as evidenced by transient co-localization, while others do not show detected temporal and physical overlap with RNAPII. We also have extended the quantitative analysis, adding simple correlation analyses (Fig. 4e,f), to more broadly support the lack of persistence of dBrd4 in the RNAPII clusters (Fig. 4f). This simple and unbiased analysis supports the conclusion that dBrd4/RNAPII co-clusters do form but are not persistent. However, because there may be other as yet unrecognized steps in the maturation of transcriptional hubs, we do not believe that colocalization justifies a conclusion that direct physical interaction of dBrd4 and RNAPII is essential to the maturation of transcriptional hubs, nor can we be certain that dBrd4 clusters that showed no overlap in time and space with RNAPII might not operate through an unrecognized intermediate to promote later recruitment of RNAPII.

Finally, the quantification of cluster intensity is required in Fig 1, Fig 6 and Fig Supp 7 because it is very difficult to judge. Also, the quantification of the mNeonGreen-Zld would help very much as the color code might not clearly show differences (why not using black and white instead?).

While not fully specified by the reviewer, we have provided the quantification of clustering using variance for the following experiments.

- mNeonGreen-Zld in the control and dCBP RNAi embryos
- Zld, dBrd4, and RNAPII in embryos injected with water or alpha-amanitin

We also replaced colored images with grayscale ones wherever possible, hopefully making the contrast more obvious.

2 – Concerning the genetics, the dBrd4- context correspond to the Jabba trap cytoplasmic sequestering using the Bcd-3'UTR to express the sequestering Jabba only after fertilization. As the Bcd3'UTR also anchored the RNA at the anterior pole, it is likely that the Jappa Trap mRNA will be anchored at the anterior pole and therefore that the dBrd4 sequestering will be very efficient around the anterior pole and less efficient in the middle of the embryo. It will be important to show it and to quantify the staining of Rbp1 shown in Figure 3d. It is also possible that the version of the Bcd-3'UTR used only control expression after fertilization but not the anchoring of the RNA at the anterior pole, but if this is the case, it has to be mentioned in the manuscript.

The reviewer is correct that the *bicoid* 3'UTR should sequester the maternally expressed mRNA encoding JabbaTrap at the anterior pole, in addition to inhibiting its translation until fertilization. We have tested

this JabbaTrap approach using *Mat-tub-Gal4* and *UASp-JabbaTrap-bcd3'UTR* on various GFP-tagged targets. We found that when the target is more abundant (e.g. EGFP-Rpb3, unpublished data), we only observed complete trapping for the nuclei near the anterior pole. When the target is expressed at lower levels, as in the case of dBrd4, we observed complete trapping for all the nuclei throughout the embryo. These results indicate that the JabbaTrap protein deployed by this approach is indeed enriched near the anterior pole, but either a lower level of JabbaTrap is also present away from the anterior pole or the target protein can diffuse from distant locations to be effectively trapped by the more localized JabbaTrap. We have provided the representative images and mentioned this detail in the corresponding Results section.

Also, it is very difficult to follow the exact genotype of the embryos analyzed. These needs to be indicated in the Figure legend or in the Material & Methods or supplementary Table 1. For instance, in Figure 1a, the embryos injected were likely “w, mCherry-Rbp1, sfGFP-Zld” and in Figure 1c, the embryos were likely from a cross between females *Mat-tub-Gal4* with males carrying the shRNA concerned but it is not clear how many copies of the EGFP-Rpb3 transgene they carry.

We have described the crosses in more detail in the Methods section and provided the maternal genotypes in Supplementary Table 2.

3 – The authors observe the spatial separation of the dBrd4 and RNAPII clusters (Figure 5). This observation is interesting even though a bit surprising because in principle, the distance between the two complexes at the expected activated loci is supposed to be below the resolution limit of the imaging. Nevertheless, the quantification shown in Figure 5a and 5b are very appealing. As mentioned already above, quantifications on many clusters are necessary to support the claims. This might also include to work on the temporal issue to determine if the spatial separation is observed during the whole period when the co-clustering occurs. Also, since the authors have used the MS2 reporter, it is important to determine if a similar observation is made between dBrd4 and the transcribing loci (MS2) and between PolII and the transcribing loci (MS2).

First of all, we would like to clarify the distinctions between the clusters we visualized and the DNA-bound complexes. Given that dBrd4 and RNAPII clusters are dependent on Zelda and later become sites of active transcription, their formation apparently involves specific interactions with DNA. Nonetheless, each cluster is bright and so contains many copies of the tagged protein, presumably assembled through protein-protein interaction that leads to oligomerization or liquid-liquid phase separation. The large sizes of the clusters, which allow us to resolve them by confocal microscopy, also mean that each complex if formed by direct interaction with DNA would involve a large span of DNA. The spatial separation of dBrd4 and RNAPII clusters indicates that the two do not persist homogeneously comingled, suggesting either a model in which dBrd4 and RNAPII first comeingle and then undergo a de-mixing process, or a model that the surface of the dBrd4 clusters nucleates the assembly the RNAPII clusters. In other words, we do not suggest that we are resolving the few dBrd4 and RNAPII proteins that bind adjacently on the DNA, which might be below the resolution of the light microscopy, although many authors have reported resolution of enhancer regions from promoters. To further support our conclusion, we have quantified the distance between dBrd4 and RNAPII clusters as well as that between RNAPII subunits as control. We have now described in more detail the apparent overlap in dBrd4 clusters and RNAPII clusters. Some RNAPII joins some dBrd4 clusters, and initially the RNAPII signal appears coextensive with the dBrd4 signal, but this changes. As we now describe in more detail, RNAPII clusters do not retain dBrd4 signal. Following multiple c-clusters during the maturation phase suggests different rearrangements. At some co-clusters, the dBrd4 signal progressively fades from RNAPII at some clusters, while other clusters seem to undergo a rapid physical disruption and de-mix/separate progressively. We are uncertain whether the variability is a result of catching a kinetic process at different points or whether different clusters mature differently, but in all cases the result is the loss, or at least severe reduction, of dBrd4 from the RNAPII clusters.

The separation of RNAPII clusters from MS2 signal for the *hunchback* reporter has been previously reported (Cho et al., Cell Rep., 2022; see its Fig. 4).

We have attempted to perform simultaneous live imaging on dBrd4 and MS2. However, we rarely observed their colocalization. It is likely due to the delay between dBrd4 clustering and active transcription. That is, although some dBrd4 clusters temporally overlap RNAPII clusters, the colocalized dBrd4 either disperses or resolves from RNAPII clusters before the detection of MS2 signal. Consequently, our failure to detect overlap of dBrd4 and MS2 signals could be the result of a lack of overlap in time, and dBrd4 may well have been present at the same locus as the MS2 signal.

4 – To detect a transcribing locus, the authors use a MS2 reporter for the *hunchback* gene which carries the MS2 cassette in 5' of the transcribed sequence. This position of the MS2 cassette allows to detect the transcribing RNAPII as soon as they elongate, and the fluorescent signal accumulates very easily and very rapidly at the transcribed locus. This signal is however so strong that the dynamic range of the fluctuation does not allow to catch easily the OFF-time of the promoter. It is therefore not very surprising that the locus is still showing MS2 signal while it is not anymore in an RNAPII cluster. This must be explained properly in the text.

We are unsure of what is being suggested here. It should be noted that any nascent transcript generating MS2 signal should also be accompanied by elongating RNAPII. Even during the promoter OFF state, there should be previously initiated RNAPII transcribing the body of the gene – the transcription unit – until all elongating transcripts have completed. We reported that the visible RNAPII cluster declines in intensity while the MS2 signal is still increasing. Since the MS2 signal is increasing, the signal is not yet saturated and new polymerases must be loading on the gene (at least recently enough to newly transcribe the 5' MS2 repeats). This discordance of one signal going down while the other is going up means that the RNAPII clusters we are visualizing (signal going down) are not composed exclusively of transcribing RNAPII (signal going up), because each nascent transcript is accompanied by an RNAPII. As we previously reported (Cho et al., Cell Reports 2022; see figure 2 and associated text), the disappearance of RNAPII clusters when MS2 signal is still rising indicates that the pool of transcribing RNAPII is much smaller or more disperse than the pool that forms the visible clusters. This finding is also apparent but less clear in the control panel of Fig. 7f of this manuscript. Consistent with previous findings (Darzacq et al., 2007), we proposed a model (Cho et al., 2022) that an excess of RNAPII is recruited to a locus in advance of transcription, while only a minority of the RNAPII in this pool initiates from the promoter and is converted to an elongating form. We have revised the manuscript and further illustrated our model in the new Figure 8. While this hopefully makes clear the distinction between promoter-associated RNAPII clusters and gene-body-associated transcribing RNAPII, we hope that curious readers will consult the prior paper to see the details.

Minor points :

1 - The authors should indicate somewhere for each of the stills if they correspond to maximal z-projections (it is only mentioned in Figure 2a legend) or if they are just one z-stack.

We have included this information in the figure legends.

2 – It is not clear why the curves shown in Fig 2b, Fig 2d, Fig 4b, Supp Fig 4 are only starting at 20 or 30 seconds while the signals are above 0.

For Fig. 2b,d (now in Fig. 3b and 4b) and Supp Fig. 4 (now removed), we defined the 0-second time point as the exit from mitosis. The signals obtained upon the initial exit from mitosis reflect the proportion of proteins retained in the nucleus along with the chromatin (dBrd4), or those imported upon closure of the nuclear envelop (RNAPII), or both (Zld). We chose not to quantify the 0-second frames because the intensities in the nuclei are still faint (Rpb1) or highly heterogenous (dBrd4) and because signals are unrelated to the new formation of clusters, which mostly occurs after 30 seconds. We have relabeled the x-axis in these figures to indicate what 0-second time point means.

For Fig. 4b (now Fig. 4d), we have relabeled the time and x-axis to indicate the earliest detection of RNAPII clusters.

As explained above, the method we used to quantify clustering also detects background noise. Therefore, even when there are no prominent clusters, the heterogeneity in the nuclear background would still produce a baseline variance. We have explained this more clearly in the manuscript.

3 – The sentence line 136 – line 137 : “Additional evidence...” is not clear and not useful for the rest of the manuscript.

We have removed this sentence as suggested by the reviewer.

4 – The term “chromatin-bound” (line 161) must be explained. No experiment is really showing that the clusters detected are bound to chromatin.

We thank the reviewer for pointing out the problem with the term “chromatin-bound”. What we meant to be referring to was the strong foci of dBrd4 associated with mitotic chromosomes. It is these foci that rapidly disperse upon entry into a new cell cycle. Both decompaction of mitotic chromosomes and dispersal of the foci might contribute to this decline. In any case, the presentation of the localization of dBrd4 has been re-written to avoid apparent unsupported claims about chromatin-bound dBrd4.

4 – The transgene allowing the expression of the MCP-mCherry is not referenced in the Supplementary Table 1.

We have added the fly lines to Supplementary Table S1.

Comments from Reviewer #2:

This study by Cho and O’Farrell uses high-resolution live imaging of tagged proteins to visualize the transcriptional machinery during zygotic genome activation in the early fly embryo. The authors demonstrate that the pioneer transcription factor Zelda, the acetyltransferase dCBP, and the reader BET protein dBRD4 form nuclear clusters / hubs during interphase of early nuclear cycles. By assessing the consequences of knockdown of the proteins, the authors further show that Zelda works with dCBP to nucleate dBRD4 clusters, which are necessary for the formation of RNAPII hubs. This study advances our knowledge of the temporal order of events that lead to transcriptional initiation. I would recommend publication after the following questions and concerns are addressed:

We thank the reviewer for the opportunity to revise our manuscript and respond to the reviewers’ concerns.

1) It is difficult to tell if Zld and Rpb1 show very little overlap from one z-plane image. In Fig 1a, the three most distinct and brightest Zld clusters (150-210, bottom left) actually do overlap with the three most distinct Rpb1 clusters. Has a quantification been done of the number of clusters for each protein and then how many of those overlap for each time point? In other words, is it possible to put a number on “rarely colocalized?”

The reviewer clearly examined the data carefully and is correct. However, as explained above to reviewer #1, there is technical difficulty in identifying clusters especially for Zld, which shows highly heterogenous clustering in terms of sizes and intensity that change too rapidly for us to track. In the revised manuscript, we explained more clearly that, given the abundance of Zld clusters, some will randomly overlap, and we can’t be certain that a Zld focus seen in a similar location in two frames of movie are actually the same focus. We have revised the presentation to emphasize the difficulties and focus attention on the interpretation that we feel is definitive: Zld does not persist in RNAPII clusters that form. We also added a brief discussion of the apparent coincidence that this reviewer noted. If one follows the time course in Figure 1b, there are peculiarities in the association of the Zld signal with the RNAPII signal. The RNAPII foci first appear adjacent to, but not coextensive with, the Zld signal. In the following frames, there were

pretty dramatic shifts in the relative positions of the two signals as the RNAPII signal intensifies and the Zld signal declines. Before the Zld signal disappears, there appear to be small Zld foci with surrounding RNAPII signal. This tantalizing sequence might provide important clues of the relationship of Zld to the formation of RNAPII clusters, but it is not easily consistent with direct recruitment by association, and, in the extreme, could be dismissed as random events in a crowded field of foci that is rapidly changing/moving. We do see additional events not unlike those shown (and now discussed), and we do not dismiss the possibility of transient overlap. We note that the observations under discussion here do fit our conclusion, which we hope is now more clearly emphasized in the presentation: there is not persistent association of Zld in the RNAPII clusters that form. This lack of persistence is inconsistent with common descriptions of cooperative interactions among components assembling a mixed complex, and the possible existence of transient interactions does not contradict our conclusion.

2) Line 174. Here the authors make note of the observed overlap of Brd4 and Zld clusters, but from looking at Fig. 2e, there are only a few instances of obvious overlap, similar to Zld and Rbp1. Since there is no rigorous quantification of clusters in either experiment, it is difficult to conclude if there is more overlap between Zld and Brd4 than Zld and Rbp1.

When we discovered an intermediary (dBrd4) in the causal sequence of RNAPII cluster formation, we were expecting to see more overlap between the earlier (Zld) and the later (RNAPII itself) components; perhaps the reader will have similar expectations. However, for Zld and dBrd4 this is not really obvious, and we have not concluded that physical overlap between Zld and dBrd4 is greater than that between Zld and RNAPII. Similar to our response to point (1), Zld clusters are difficult to identify, preventing us from quantifying their overlap with either dBrd4 or Rbp1. Since we find that CBP is required for Zld action in this pathway, we suggest that local histone acetylation by CBP marks sites of prior transient Zld association events to mediate dBrd4 recruitment. As noted above, we have revised our manuscript to emphasize the observation that Zld intensity is low or absent in matured dBrd4 or Rbp1 clusters. Additionally, we do affirm that our data show a temporal progression in which the factors acting upstream in a causal cascade form foci earlier than those that are downstream. Further, we note that, because the number of dBrd4 foci and RNAPII clusters are less than the Zld clusters, they are easier to track, and we do see transient coextensive overlap of dBrd4 foci with forming RNAPII clusters that can be seen in a few successive frames. Real-time imaging in the earlier cycle (cycle 11) when there are fewer foci shows convincing evidence for at least transient associations between dBrd4 and RNAPII clusters (see added Supplementary Movie 1), but we remain cautious about drawing functional conclusions regarding the possible role of direct physical interactions between the two proteins. We hope revisions to the presentation and new representative images are clearer in showing these aspects of our data.

3) Line 79. The authors appear to be hypothesizing that the lack of visual colocalization between Zld and Rbp1 is because the clusters are not only transient, but dependent on a specific temporal order of events such that you would not expect them to colocalize. It is known from super resolution imaging (Mir and Eisen) that Zld binds very transiently to its target sequences, so this hypothesis is plausible even if there were no intermediate steps. Was a more precise tracking analysis similar to the cluster tracking done for dBrd4 and Rbp1 (in Fig. 4) done for Zld and Rbp1 to rule out colocalization?

As explained above, Zld clusters are numerous and so some overlap with Rbp1 clusters can occur randomly. Currently, we do not yet know whether any of those transient and infrequent colocalization is functionally important. Notably, our imaging rate is much slower than that of Mir and Eisen, and as mentioned in the review, the very short lifetimes of Zld foci described by Mir and Eisen mean that it is possible that transient colocalization occurs at much shorter time scale. We now make it clear that we are not ruling out transient colocalization. Rather, our data show an absence of persistent colocalization. Also, we argue that the existence of an intermediate step is demonstrated by genetic perturbation, and there might be additional inputs from, for example other TFs. We prefer the notion that very transient Zld interactions, perhaps boosted by other TFs, influence dCBP-mediated histone acetylation, and the

resulting accumulation of histone acetylation integrates Zld/TF inputs to reach a threshold that triggers dBrd4 assembly into clusters.

4) Lines 133. Is there evidence that the Zld clusters actually represent Zld bound to its target motifs? Was this shown here or elsewhere? If so, please show or reference. If not, the suggestion “Zld presumably at its target DNA sites” is overstated.

We thank the reviewer for this critique. To our knowledge, there is still no evidence in the literature that Zld clusters are associated with its target DNA sites. We have rewritten the sentences to take out the inferred association.

5) Line 251. Where is the data that shows dBrd4 is “redistributed” to the HLBs? How was this measured? At the very least, I do not see a quantification of the signal intensity of the HLBs over time. From simple glance at the spots, they don’t appear to get bigger.

Our data show that, following the injection of alpha-amanitin, the dBrd4 clusters at HLBs became especially prominent while the rest of dBrd4 clusters were reduced. However, the reviewer is correct that this observation alone is not sufficient to support that there is a redistribution of dBrd4 from non-HLB clusters to HLBs. We have revised the text accordingly to avoid making this claim.

6) Line 88. Does transcriptional bursting occur in early nuclear cycles? Seems nc11-12 are too short for multiple bursts to occur, never mind be visualized.

The reviewer is correct that nc11-12 are so short that, if there was bursting, it would have to be so rapid that we would likely have missed it in our analyses. The best data to address the reviewer’s question comes from EM data visualizing the arrays (Christmas trees) of nascent transcripts in embryonic chromatin spread (e.g McKnight and Miller, Cell, 1976; 1977; 1979). In cycle 14 when longer transcription units are expressed (1979), they observed interruptions in these arrays that represent transient interruption in initiation. Notably, these EM data are best at detecting short bursts that are less than the elongation time. Such short bursting events could occur during the early syncytial cycles, but in their analysis of the early cycles (1976, 1977), these authors reported only densely packed short transcription arrays without gaps. Thus, if short bursts do occur during nc11-12, they are likely rare, and we can mostly consider transcription confined by the short interphase window to be a single burst.

The model we are proposing includes two components, i.e. the mechanisms that initiate a burst and those that attenuate it. We suggest that the maturation of transcriptional hubs are similarly involved in initiating a burst in both the shorter and longer cycles. However, different mechanisms might be responsible for attenuating the burst. In the shorter nc11-12, entry into mitosis and consequently the abortion of transcription likely plays a major role. In the longer cycles such as nc13, the attenuation can occur independently of mitosis. Our results from the abrupt injection of alpha-amanitin in early nc13 suggest that a prolonged period of transcription contributes to the dispersal of transcriptional hubs. This leads us to think that, in the subsequent and much longer cycles, this feedback attenuation of transcriptional hubs will be more important so that assembly and dispersal of transcriptional hubs will make direct contribution to bursting.

7) Fig. 3. Please label frames in figure since in the text you refer to “frames” versus seconds.

We have labeled Fig. 3b (now Fig. 2a) more clearly and revised the text accordingly.

Comments from Reviewer #3:

Review: Cho and O’Farrell, ‘Stepwise modifications of transcriptional hubs link pioneer factor activity to a burst of transcription’

In this manuscript, the authors investigate the formation and impact of transcriptional hubs. They address this important question in the early *Drosophila* embryo, amenable to live imaging and genetic

manipulations. In particular, they examine the spatio-temporal clustering of major transcriptional regulators such as Pol II (Rbp1 and Rbp3), the acetyltransferase dCBP, the transcriptional co-activator dBrd4 and the pioneer factor Zelda, by utilizing through live imaging from endogenously tagged loci. The key finding of this study is the sequential formation of transcriptional hubs constituted by these factors as well as their cascade of dependencies. Except for HLB loci (which appear to be controlled by an alternative cascade), Zelda clusters seem to come first, working through co-activators like dBrd4 and dCBP to then elicit the formation of Pol II clusters. Timing and dependencies are clearly demonstrated via elegant genetic manipulations (RNAi or Jabba trap + live imaging). The manuscript ends with the surprising finding that transcription feedbacks on a subset of transcriptional hubs, namely dBrd4 and Pol II clusters, in order to promote their dispersal.

In general, I think the findings of this paper are original, well-presented and constitute an important contribution to the field of gene regulation during development. However, the manuscript lacks quantifications to back up most of the conclusions. While I am convinced by the findings regarding the transcriptional hub dynamics (Fig1 to 5), I am a bit more skeptical about the last part of the manuscript regarding the effect of sustained transcription on clustering (see below point 2).

We thank the reviewer for acknowledging the strength of our approach and the importance of our findings. We have provided additional quantifications and statistical analysis to support our conclusions. Regarding the reviewer's concern about the effect of sustained transcription on clustering, we responded in more detail below.

Major comments:

1. Most of the results are descriptive and not quantified. The authors generally mention in the figure legend that the results were observed in a number x of embryos, but without extracting quantitative information from these embryos.

Since the data have already been obtained and since the procedure for image analysis is already in place (for example Fig 2b), it would be relatively easy to strengthen each figure with the appropriate quantifications.

General number concerning hub kinetics could be extracted from the data to learn: how fast do hubs form? how fast do they dissolve? how long do they interact with DNA? how long do they interact with a given TF/GTF?

We have substantially revised our manuscript and provided quantifications wherever possible. As for the suggestion to extract hub kinetics, we agree that those will be extremely informative and useful, but at present we are unable to track Zld clusters, and while we are able to track some dBrd4 and RNAPII clusters, we cannot reliably follow all of them. As described in the responses to reviewer #1 and 2, this limitation prevents a thorough characterization of hub kinetics. We hope that with application of more powerful microscopic techniques and fluorescent probes, we will be able to answer these questions with confidence.

Here are a few examples of findings that would be solidified by quantifications:
-Fig1a, bottom: signal at HLB clusters seems not to be affected by Zelda depletion.

The effect of Zelda depletion on the recruitment of RNAPII to HLBs has been carefully examined by Huang et al., 2021 (PMID: 34614388). The authors concluded that RNAPII accumulates more abundantly at HLBs when Zelda is knocked down. Our data in Fig. 1a failed to detect apparent increase in RNAPII recruitment to HLB following trapping of Zelda, but certainly RNAPII is still robustly recruited to HLBs following Zld depletion. Since our study focuses on the regulation of non-HLB clusters, we decided not to perform and include the quantification suggested by the reviewer in this manuscript and note that we intend to follow-up with further studies focusing on the cascade operating at HLBs.

-Fig1b/line 116: RNAP II clusters emerged, but infrequently overlapped with Zelda cluster.

As explained above, Zelda clusters are numerous and vary substantially in their intensity and sizes. Also, work from Mir and Eisen achieved much high imaging rates and revealed that Zelda clusters have very short half-lives, considerably shorter than our frame rate. This prevents us from tracking Zelda clusters from frame to frame in our imaging—indeed it is uncertain whether there are meaningful long lived Zld clusters. This compromises our ability to distinguish coincidence from co-clustering. We have revised the manuscript to describe our observations more clearly. And, as discussed above, because of the difficulty assigning significance to occasional and transient overlap of foci, we have altered the presentation to reduce the emphasis on whether or not there are transient associations. Instead, we emphasize the point and the conclusion that the earlier acting regulators do not persist in the RNAPII clusters as they mature.

-Fig 2e.f/line 173: visual inspectionnewly formed dBrd4 clusters occasionally colocalized with Zelda clusters....Maybe quantify % of coloc at different time points.

As discussed above and in responses to other reviewers, due to the difficulty in defining and tracking Zelda clusters, we cannot say whether rare observations of colocalization with dBrd4 are significant or due to chance. We have revised the manuscript to emphasize our conclusion that Zld does not persist in matured dBrd4 clusters and to directly state that we cannot address whether there are significant direct associations of Zelda and dBrd4 clusters. However, our data do document a temporal sequence in which Zld clusters precede the formation of dBrd4 clusters.

-Figure 4: temporal disconnect between dBRD4 and RNAPII clusters. The authors could display the distribution of 'delta-t', lag time between peak signal for each cluster and envisage correlation analyses.

Since it is difficult to track the clusters even manually, we took a different approach and asked whether the correlation between dBrd4 and RNAPII intensity changes over time. As shown in Fig. 4e and 4f, when RNAPII clusters first emerged, there is a moderately positive correlation between dBrd4 and RNAPII intensity. This correlation diminishes later on, supporting the earlier dispersal of dBrd4 from the hubs.

-Figure 5: same point as before, for spatial disconnect: quantify distance between clusters to observe the frequency of colocalization.

We have quantified the distance between co-clusters for dBrd4/Rpb1 and Rpb3/Rpb1.

-Figure 6a/line 239: dBrd4 clusters become more intense. Quantify size and intensities. -Figure 6b/line 258: MCP TS foci: quantify over time.

We have provided these quantifications of changes in cluster intensity – now figure 7.

2-Conclusions regarding the impact of a sustained transcription on cluster decay are not solidly demonstrated.

-The inhibition of transcription with alpha-amanitin injection is difficult to interpret, as the timing of action of this drug is poorly characterized in the fly embryo. In mammalian cells the delay between drug application and effect can require several hours. The fly embryo is uncellularized but the large volume the drug has to diffuse into remains a challenge. Here, MCP-MS2 dots are still observed minutes after drug injection, indicating transcription did not arrest. Alternatively, the remaining spot corresponds to Pol II blocked with MS2-nascent mRNAs, exhibiting high retention of MCP. One can decipher this by fixing embryos after drug injection and performing smFISH on MS2 in the absence of MCP detector protein or perform FRAP in live MS2/MCP embryos.

We respectfully disagree with the reviewer for the following reasons. First, the long delay between alpha-amanitin treatment and its effect in mammalian cells is known to be due to the low membrane permeability of the drug. In our case, the drug was directly injected into the embryo, so the inhibition of transcription should be fast if not immediate after injection. Indeed, dramatic and immediate (within the context of our manipulations) effects of drug injection were observed. Second, the diffusion of the drug in the embryo is also fast, as even larger molecules such as fluorescent proteins spread readily to a large area shortly after injection (see Movie S1 in Yuan and O'Farrell, 2016 for example. PMID: 26915820). In addition, we

made sure to record the area closest to the injection site, so the time for the drug to diffuse across the entire embryo should not be a significant concern in our experiments. Third, the observed MCP foci do not argue against the inhibition of transcription, as we injected the drug when transcription just began. As the reviewer recognized, MCP foci could persist without growing if alpha-amanitin blocked RNAPII translocation without causing it to abort. We have further provided quantification of changes in the intensity of MCP to support this conclusion. Finally, the major conclusion we are drawing is the impact of the drug on RNAPII clusters, and we observed stabilization of the clusters in embryos without the MS2/MCP systems. Taken together, we believe that our data firmly support both the expected effect of alpha-amanitin and our conclusion. Therefore, we see no requirement to perform these experiments suggested by the reviewer to present our findings.

-Authors should perturb transcription with alternative methods: genetically (RNAi of overexpression of transcription initiation/elongation factors) or with other drugs (flavopiridol, triptolide, DRB).

Since transcription appears to have dual roles in regulating RNAPII clusters, initially promoting them and later destabilizing them, we need alternative methods that can be abruptly introduced shortly after cluster formation. It is thus not possible to use the genetic approach.

As for the suggestion to use other drugs, we want to point out that the exact feedback mechanisms by which transcription regulates clusters of dBrd4 and RNAPII remain unclear. Since different drugs target different steps in the transcription cycle, from initiation (triptolide), pause release (DRB and flavopiridol), to elongation (alpha-amanitin), these drugs might have different impacts on clustering. Indeed, our preliminary experiments using triptolide showed that its abrupt injection in early nc13 failed to arrest MS2 reporter expression or to stabilize RNAPII clusters. We wish to investigate these questions more thoroughly in a future study, including plans to perform JabbaTrap on Cdk7 and Cdk9. We do agree that a more refined understanding of the effect of transcription on the stability of transcriptional hubs is important, but for this manuscript, we argue that the experiments using alpha-amanitin are well controlled, and our conclusion that transcription plays some role in regulating hub dynamics is firmly supported and worthy of being reported.

-The observations are made at stages pre-ZGA, before the major transcription wave. It would be interesting to analyze how Brd4 and Pol II clusters behave during nc14, and whether they would tend to be more disperse.

We are unsure of the question that the reviewer would like to address. We recognize two issues that would be of interest to explore in cycle 14. As brought up by reviewer #2, the very short window of transcriptional opportunity in the early cycles leaves little time for repeated rounds of transcriptional bursting. Furthermore, the number of genes that are expressed in the early wave might be limited to those that can be activated in the remarkably short time available following mitotic exit. These are questions that we intend to investigate in the future. Our preliminary observations in the much longer cycle 14 suggest that the early post-mitotic events are very similar to those in the earlier cycles (see control embryo in Figure 2d). While it will be interesting to extend our studies to later cycles, we do not think that the observations and conclusions we present depend on such an extension.

3-Since Zelda, Pol II, dCBP and dBrd4 Chip-seq results are available for early Drosophila embryos, it would be interesting to discuss the temporal cascade discovered in this study with the overlap of bound targets.

More generally, the 'DNA' content of these clusters is not discussed in the manuscript, yet it represents an important piece of the puzzle.

We agree with the reviewer that the DNA content of the clusters remains incompletely understood and note that visual detection of protein clustering offers an experimental perspective that is different than that derived from ChIP-seq, which examines only cross-linkable protein-DNA/chromatin interactions. "Overlap" of two proteins in ChIP-seq data does not mean that the two proteins co-localize—but only that

they interact with nearby sites in the genome. Given the way ChIP-seq experiments are done, the two proteins that bind to similar regions do not have to be in the same cell, or present at the same time in the cell cycle. In fact, they might even compete for the same site and never bind together even when they are present and active together. Therefore, an overlap in ChIP-seq has a different meaning than in visual colocalization of clusters. In addition, to our knowledge, the ChIP-seq data for dBrd4 and dCBP in blastoderm embryos are still unavailable. Thus, even though the issues underlying the suggested analysis are of interest, we were unable to perform this work and feel that new experimental approaches will be needed to explore the relationship of ChIP-seq data to visualization of protein clustering in the nucleus.

We emphasize that the central findings in this manuscript are the cascades of dependency and sequential transformation of transcriptional hubs. Elucidating the associated DNA content will be an important next step in the future.

4-Discussion regarding the implication of the sequential formation/dissociation of clusters with respect to transcriptional bursting (line 276). I would nuance the statement, as the authors only monitor a specific timescale of transcriptional dynamics. We now know that bursting is multiscale (Lammers COGD), ranging from seconds to days. Thus TF/GTF/co-activator clustering may impact a specific timescale of transcriptional bursting but cannot be the sole source of bursting.

While we documented the sequential formation and dissociation of the hubs at the timescale of minutes in the early embryo, we argue that the speed of this progressive process can be easily tuned by changing regulation at individual steps to give rise to bursting at longer or shorter timescales. In line with this idea, initiation of DNA replication depends on the same machinery, but the temporal program of DNA replication is dramatically extended during development.

Even though we argue that the cluster dynamics can provide a mechanism for transcriptional bursting at a large range of timescales, we agree that we cannot rule out other mechanisms that contribute to bursting. We have revised the manuscript accordingly.

Minor comments:

-A figure with a model would be appreciated.

We have provided such a model figure - Figure 8.

-the authors use clusters or hubs. Is there a conceptual difference?

Both words are associated with the similar concept that factors are locally enriched in subnuclear regions, but they were used in slightly different contexts. The word "cluster" was used to describe the behaviors of individual factors that we observed, while the word "hub" was used in a broader sense to refer to the subnuclear regions associated with activating genes. We have revised the manuscript to make sure that these terms are used appropriately.

- Why JabbaTrap is used via RNA injection for depleting Zelda and with genetics (UAS- JabbaTrap-bcd3'UTR)?

-The efficiency of depletion of Zelda with Jabba trap is not shown. Could the author back up the requirement of Zelda to initiate RNAPII clustering with Zelda RNAi approach?

Different methods of expressing JabbaTrap have different advantages. Injection of mRNA allows abrupt inhibition at different nuclear cycles, but the effects are usually more local and restricted to the nuclei near injection sites. The genetic approach by maternal Gal4/UASp system can achieve more complete inhibition in the early embryo, but the timing cannot be tuned.

Since the sequestration of Zelda is shown in Figure S1, we assume that the reviewer is asking for evidence of dBrd4 sequestration by JabbaTrap. As also requested by reviewer #1, we have provided the representative images of dBrd4 trapping in Supplementary Figure 5a.

We devised a modified genetic approach to knockdown dBrd4 by JabbaTrap, taking advantage of its thoroughness while avoiding the sterility cause by an absence of dBrd4 function in the female germline. The result was very effective. While we could potentially back up the findings with injection of JabbaTrap mRNA, the availability of JQ1, an inhibitor of dBrd4 binding to its acetylated targets, provided a second means of testing the role of dBrd4 in RNAPII clustering. Furthermore, the inhibitor provides additional insights into its actions. As shown in Supplementary Fig. 5b, the injection of JQ1 significantly reduced the chromatin binding and clustering of dBrd4, and the clustering of RNAPII at non-histone genes was reduced. This experiment not only confirms the role of dBrd4 in mediating RNAPII clustering but implicates recruitment and localization of dBrd4 by interaction with acetylated targets in the process. Furthermore, this injection strategy allows us to abruptly suppress the dBrd4 input at syncytial stages.

-Line:628 explanations of how the quantification was performed is unclear: what is a boundary? How was it defined? What is the zero in the x axis?

We have added dashed lines to Figure 5 to indicate the ROIs used for quantification. The lines were drawn manually based on visual inspection of the data.

-to be rigorous, the genotype of embryos depleted for dBrd4 or Zelda with the Jabba trap trick should not be named 'dBrd4-' or 'Zelda-'.

We have relabeled the control and JabbaTrap embryos in the figures following the same style we used in the original paper reporting the JabbaTrap method (Seller et al., 2019. PMID: 30808658).

-line 131: why shifting to Rbp3 while the rest of the figure employed Rbp1?

Since EGFP-Rpb3 and mCherry-Rpb1 have been shown to colocalize with each other and behave similarly (Cho et al., 2022. PMID: 36261005 and Figure 5 of this manuscript), we used them interchangeably as reporters for RNAPII in our experiments.

In the JabbaTrap experiments, mCherry-Rpb1 was used so that it would not be sequestered by the anti-GFP nanobody in the JabbaTrap. In the RNAi experiments, EGFP-Rpb3 was used to provide a better signal and to confirm the JabbaTrap result.

-Endogenous tagging of dBrd4 can be difficult (and the authors mention some issues in the methods: line 350). The authors claim that embryos laid by sfGFP/Halo-dBrd4 mothers have a similar hatching rate than controls (line150). Could the authors show the hatching rate quantification in a supplementary figure?

We did not have too much difficulty tagging the endogenous dBrd4, and multiple lines were obtained for each construct. The cell-cycle defect in embryos expressing HaloTag-dBrd4 was only observed after injection of the HaloTag TMR ligand, suggesting that the ligand labeling, rather than the protein tag, is the major issue. We have provided the hatch rates in Supplementary Figure 3.

-Figure 3d: in dBrd4-, in nc14, nuclei seem to show only 1 HLB cluster. Is this true? Can the author comment on this result (if true)?

Pairing of homologous chromosomes is a well-known feature of mitotic cells in *Drosophila*, but this pairing only emerges as development proceeds. Previous work had described the increase in pairing as marked by the onset of fusion of HLBs which became prevalent in cycle 14 (Hiraoka et al., 1993. PMID: 8425892). While Fig. 3d (now Fig. 2d) was presented to show the gastrulation defects following dBrd4 knockdown, it also shows the progression of HLB pairing. Though somewhat obscured by the strong early signals from none HLB loci, the control embryo shows that most nuclei have two unpaired HLBs at the beginning of cycle 14 (upper-left panel). In contrast, many nuclei exhibit only one HLB in the later stage following cephalic furrow formation (upper-right panel). Similarly, at the later stage, the dBrd4 knockdown embryo (bottom-right panel) has many nuclei with apparently a single HLB. The reviewer expressed an interest particularly in this effect in the experimental embryo. Because the dBrd4 knockdown embryo is highly abnormal at this stage, we cannot comment in any detail about the role of dBrd4 in the pairing phenomenon except to say that it still seems to occur following knockdown.

-missing references:

The mitotic retention of dBrd4 is published in and should be cited (Ringrose lab).

To our knowledge, the data on the mitotic retention of dBrd4 from the Ringrose lab is still unpublished. We are thus unable to cite their parallel findings.

The effect of Zelda on enhancer hub formation is analyzed in Espinola et al and not discussed in the manuscript.

We have cited this paper by Espinola et al. when discussing the potential function of transcriptional hubs in mediating enhancer hub formation.

REVIEWERS' COMMENTS

Reviewer #1 (Remarks to the Author):

The authors answered most of my concerns and I really appreciate the efforts they made to provide quantitative analysis of their data : using the variance to assess clustering in imaging data is a great idea which help very much to catch the relative temporal dynamics of cluster formation (Fig 3b and Fig 4b for instance).

Reviewer #2 (Remarks to the Author):

The authors have addressed my concerns by making substantial changes to the manuscript. These changes now clarify the issues of assessing colocalization of Zld foci with Rpb1 and dBrd4 foci due to the sheer number of Zld foci and their dynamic behavior. Given that the data do support the conclusion that there is a “temporal progression in which the factors acting upstream in a causal cascade form foci earlier than those that are downstream,” I am in favor of publication.

As an added note, I appreciated the response to my question about bursting in the early cycles, and the response citing McKnight and Miller papers using EM to visualize transcripts and the “interruptions in initiation” seen in nc14.

Reviewer #3 (Remarks to the Author):

The manuscript is significantly improved and overall I am happy that the authors have addressed my concerns. One of my comments (and that of the other reviewers) was to provide a better quantification of clusters dynamics/kinetics. The authors explain the technical challenge of following individual clusters with the slow imaging they employ.

I understand this limitation and as long the conclusions are nuanced (which is the case in the revised text), I support the publication of the revised manuscript.

Comments on two very minor points:

1-I mentioned that Bdr4 mitotic retention was published by the Ringrose lab. The authors answered that this result was still unpublished. Well, it seems that they did not look carefully at the published work I mentioned. Here is the reference and the exact quotation from that manuscript. DOI: 10.1007/s00412-021-00762-z

‘We first generated flies carrying an EGFP::FSH-S transgene and investigated its binding behaviour during mitosis and interphase as described above for ASH1 (Figure S3A-E). We observed that FSH-S is strongly enriched on mitotic chromosomes (Figure S3D,E).’

2- I also suggested to discuss the results (obtained with imaging) with genomics data, knowing of course that one is single-cell while the other concerns bulk assays. I still believe that the comparison is worth discussing. But I leave it to the authors to decide on this point.

The rebuttal mentions that they are not aware of published CBP or dBRD4 occupancy data in blastoderm embryos: Koenecke et al, 2016 (CBP) DOI 10.1186/s13059-016-1057-2

Hunt et al...Mannervik: Biorxiv, 2022 doi: <https://doi.org/10.1101/2022.10.25.513691> (CBP and dBRD4).

We would like to thank all the reviewers for carefully reviewing our work. Their valuable input has been tremendously helpful in improving the quality of the manuscript. We are pleased that the revised manuscript has addressed most of the reviewers' concerns. Below, we respond to the additional comments of reviewer #3.

Reviewer #1 (Remarks to the Author):

The authors answered most of my concerns and I really appreciate the efforts they made to provide quantitative analysis of their data : using the variance to assess clustering in imaging data is a great idea which help very much to catch the relative temporal dynamics of cluster formation (Fig 3b and Fig 4b for instance).

Reviewer #2 (Remarks to the Author):

The authors have addressed my concerns by making substantial changes to the manuscript. These changes now clarify the issues of assessing colocalization of Zld foci with Rpb1 and dBrd4 foci due to the sheer number of Zld foci and their dynamic behavior. Given that the data do support the conclusion that there is a “temporal progression in which the factors acting upstream in a causal cascade form foci earlier than those that are downstream,” I am in favor of publication.

As an added note, I appreciated the response to my question about bursting in the early cycles, and the response citing McKnight and Miller papers using EM to visualize transcripts and the “interruptions in initiation” seen in nc14.

Reviewer #3 (Remarks to the Author):

The manuscript is significantly improved and overall I am happy that the authors have addressed my concerns. One of my comments (and that of the other reviewers) was to provide a better quantification of clusters dynamics/kinetics. The authors explain the technical challenge of following individual clusters with the slow imaging they employ.

I understand this limitation and as long the conclusions are nuanced (which is the case in the revised text), I support the publication of the revised manuscript.

Comments on two very minor points:

1-I mentioned that Bdr4 mitotic retention was published by the Ringrose lab. The authors answered that this result was still unpublished. Well, it seems that they did not look carefully at the published work I mentioned. Here is the reference and the exact quotation from that manuscript. DOI: 10.1007/s00412-021-00762-z

‘We first generated flies carrying an EGFP::FSH-S transgene and investigated its binding behaviour during mitosis and interphase as described above for ASH1 (Figure S3A-E). We observed that FSH-S is strongly enriched on mitotic chromosomes (Figure S3D,E).’

We apologize for missing this paper and have acknowledged and referenced this prior work.

2- I also suggested to discuss the results (obtained with imaging) with genomics data, knowing of course that one is single-cell while the other concerns bulk assays. I still believe that the comparison is worth discussing. But I leave it to the authors to decide on this point.

The rebuttal mentions that they are not aware of published CBP or dBRD4 occupancy data in blastoderm embryos:

Koenecke et al, 2016 (CBP) DOI 10.1186/s13059-016-1057-2

Hunt et al...Mannervik: Biorxiv, 2022 doi: <https://doi.org/10.1101/2022.10.25.513691> (CBP and dBRD4).

We thank the reviewer for bringing these two very interesting papers to our attention. It was appropriate to acknowledge the Koenecke et al. paper for its demonstration of a genomic association of CBP binding with histone acetylation and at enhancers, and we have referenced it in the manuscript. However, we decided not to add a discussion of possible relationships between this and other genomic data and our imaging data, because we did not feel that it would be productive. As explained in our previous response and acknowledged by the reviewer, the ChIP-seq peaks cannot be rigorously related to protein clusters we visualized by microscopy. Indeed, ChIP signals that map the binding of two proteins to nearby genomic locations do not indicate co-residency of the two proteins at the same time and in the same place within an embryo. In contrast, our conclusions depend heavily on showing colocalization in time and space. We are performing experiments to examine the colocalization of fluorescently tagged dCBP with Zld or dBrd4, which we hope will further shed light on the molecular mechanisms activating early zygotic transcription.